

# A new paravian dinosaur from the Late Jurassic of North America supports a late acquisition of avian flight

Scott Hartman[1], Mickey Mortimer[2], William R. Wahl[3],
Dean R. Lomax[4], Jessica Lippincott[3] and David M. Lovelace[5]

[1] Department of Geoscience, University of Wisconsin-Madison, Madison, WI, USA
[2] Independent, Maple Valley, WA, USA
[3] Wyoming Dinosaur Center, Thermopolis, WY, USA
[4] School of Earth and Environmental Sciences, The University of Manchester, Manchester, UK
[5] University of Wisconsin Geology Museum, University of Wisconsin-Madison, Madison, WI, USA

## ABSTRACT

The last two decades have seen a remarkable increase in the known diversity of basal avialans and their paravian relatives. The lack of resolution in the relationships of these groups combined with attributing the behavior of specialized taxa to the base of Paraves has clouded interpretations of the origin of avialan flight. Here, we describe *Hesperornithoides miessleri* gen. et sp. nov., a new paravian theropod from the Morrison Formation (Late Jurassic) of Wyoming, USA, represented by a single adult or subadult specimen comprising a partial, well-preserved skull and postcranial skeleton. Limb proportions firmly establish *Hesperornithoides* as occupying a terrestrial, non-volant lifestyle. Our phylogenetic analysis emphasizes extensive taxonomic sampling and robust character construction, recovering the new taxon most parsimoniously as a troodontid close to *Daliansaurus*, *Xixiasaurus*, and *Sinusonasus*. Multiple alternative paravian topologies have similar degrees of support, but proposals of basal paravian archaeopterygids, avialan microraptorians, and *Rahonavis* being closer to Pygostylia than archaeopterygids or unenlagiines are strongly rejected. All parsimonious results support the hypothesis that each early paravian clade was plesiomorphically flightless, raising the possibility that avian flight originated as late as the Late Jurassic or Early Cretaceous.

# INTRODUCTION

Paravians are an important radiation of winged coelurosaurs more closely related to birds than to *Oviraptor*, that include dromaeosaurids, troodontids, unenlagiines, halszkaraptorines, and archaeopterygids in addition to derived avialans. Despite robust support for their monophyly, the interrelationships and composition of these groups remains contentious with recent studies alternatively favoring joining troodontids and dromaeosaurids as Deinonychosauria (*Hu et al., 2018*; *Lefèvre et al., 2017*; *Shen et al., 2017b*; *Godefroit et al., 2013b*; *Senter et al., 2012*; *Turner, Makovicky & Norell, 2012*),

Corresponding author
Scott Hartman, sahartman@wisc.edu

placing troodontids closer to Aves than dromaeosaurids (*Gianechini et al., 2018*; *Cau et al., 2017*; *Foth & Rauhut, 2017*; *Lee et al., 2014b*; *Foth, Tischlinger & Rauhut, 2014*; *Godefroit et al., 2013a*), joining dromaeosaurids and avialans to form Eumaniraptora to the exclusion of troodontids (*Agnolin & Novas, 2013*) or merely recovering an unresolved trichotomy between the three (*Cau, Brougham & Naish, 2015*; *Brusatte et al., 2014*).

This lack of phylogenetic resolution has been attributed to a combination of rapid rates of evolution at the base of Paraves (*Brusatte et al., 2014*) and a lack of sampling of taxa from outside the hugely prolific Late Jurassic and Early Cretaceous fossil beds of eastern China. In particular, Middle and Late Jurassic taxa *Archaeopteryx*, *Anchiornis*, *Aurornis*, *Eosinopteryx*, and *Xiaotingia* have been recovered variously as basal dromaeosaurids, basal troodontids, archaeopterygids, or non-archaeopterygid avialans (*Turner et al., 2007*; *Xu et al., 2011*; *Senter et al., 2012*; *Godefroit et al., 2013a*, *2013b*). Incomplete taxonomic sampling and unresolved relationships of basal avialans hinders tests of hypotheses for the origin of flight, and the order of acquisition of flight-associated characters in stem avians.

Here, we report a new paravian theropod, *Hesperornithoides miessleri* gen. et sp. nov., collected from the Morrison Formation near Douglas, Wyoming, USA, based on a largely complete skull and associated postcranial elements (*Lovelace, 2006*; *Wahl, 2006*). We also present a significantly expanded and updated phylogenetic analysis of maniraptorans that clarifies paravian relationships, as well as the acquisition of flight-related characters in stem avians.

# MATERIALS AND METHODS

## Specimen curation

WYDICE-DML-001 (formerly WDC DML-001) was collected on private property in 2001 (see Locality & Geologic Context below). In 2005 it was donated to the Big Horn Basin Foundation, a research and educational non-profit 501(c)3 formed in 1995 to be curated and made available for research. In 2016 the Wyoming Dinosaur Center and the Big Horn Basin Foundation merged to form a new non-profit organization, renamed The Wyoming Dinosaur Center, Inc. At that point WYDICE-DML-001 was transferred to the new non-profit, where it will be accessible to researchers in perpetuity. If the Wyoming Dinosaur Center, Inc. should ever cease to exist the donation agreement requires the specimen be transferred to the University of Wyoming's paleontological collections, ensuring it will always be available for research.

The electronic version of this article in portable document format (PDF) will represent a published work according to the International Commission on Zoological Nomenclature (ICZN), and hence the new names contained in the electronic version are effectively published under that Code from the electronic edition alone. This published work and the nomenclatural acts it contains have been registered in ZooBank, the online registration system for the ICZN. The ZooBank Life Science Identifiers (LSIDs) can be resolved and the associated information viewed through any standard web browser by appending the LSID to the prefix http://zoobank.org/. The LSID for this publication is: urn:lsid:zoobank.org:pub:6325E8D2-0AAF-4ECD-9DF2-87D73022DC93. The online version of this work

## Preparation

Multiple cycles of mechanical micro-preparation have been performed on the specimen since discovery. The specimen was collected in several blocks and later reassembled in the lab to reproduce the original in-field association. In 2004 the skull and body block were scanned at the University of Texas High-Resolution X-ray Computed Tomography Facility in Austin, Texas. Segmentation of the scan data was completed in Object Research System's Dragonfly v1.1 software to help visualize preserved elements and internal morphology. Internal cavities, such as those seen in long bones and pleurocoels were filled with the mineral barite. The high electron density of barite precluded segmentation of many of the postcranial elements due to poor visualization. During physical preparation most of the specimen was left in the minimal amount of host matrix to preserve the original association but reveal as much morphology as possible.

## SYSTEMATIC PALAEONTOLOGY

Theropoda *Marsh, 1881*

Maniraptora *Gauthier, 1986*

Paraves *Sereno, 1997*

Deinonychosauria *Colbert & Russell, 1969*

Troodontidae *Gilmore, 1924*

*Hesperornithoides miessleri* gen. et sp. nov.

**Holotype:** WYDICE-DML-001 (Wyoming Dinosaur Center, Thermopolis), a single, partially articulated skeleton consisting of most of an articulated skull and mandibles missing the anteriormost portions, hyoids, five cervical vertebrae, first dorsal vertebra, isolated anterior dorsal rib, portions of 12 caudal vertebrae, five chevrons, partial left scapula and coracoid, portions of the proximal left humerus and distal right humerus, left ulna and radius, radiale, semilunate carpal, left metacarpals I–III, manual phalanges III-2 and 3, manual unguals I, II, and III, ilial fragment, most of an incomplete femur, right and left tibiae and fibulae, left astragalus and calcaneum, portions of right and left metatarsal packets, left pedal phalanges III-1, III-2, III-3, IV-1, IV-2, IV-3, IV-4, and pedal unguals II and III and the proximal portion of IV.

**Etymology:** "Hesper," (Greek) referring to the discovery in the American West, "ornis," (Greek) for bird and "oeides," (Greek) for similar, referring to the avian-like form of derived paravians. The trivial epithet honors the Miessler family, who have been avid supporters of the project.

**Occurrence:** Douglas, Converse County, Wyoming, USA; middle portion of Morrison Formation, which has been variously dated between Oxfordian and Tithonian in age

(*Trujillo, 2006*; *Trujillo et al., 2014*), associated vertebrate fossils include the sauropod *Supersaurus*, a stegosaurid plate, and isolated large theropod teeth.

**Diagnosis:** A paravian with the following derived characters: pneumatic jugal (also in *Zanabazar* and some eudromaeosaurs among maniraptorans); short posterior lacrimal process (<15% of ventral process length, measured from internal corner; also present in *Zanabazar*, *Archaeopteryx*, and *Epidexipteryx*); quadrate forms part of lateral margin of paraquadrate foramen; small external mandibular fenestra (<12% of mandibular length; also in *Zhenyuanlong* and *Dromaeosaurus* among non-avian paravians); humeral entepicondyle >15% of distal humeral width (also in some avialans); manual ungual III subequal in size to ungual II (also in *Daliansaurus*, IGM 100/44 and *Mahakala*); mediodistal corner of tibia exposed anteriorly (also in *Archaeopteryx* and *Jeholornis*).

**Locality and Geologic Context:** In the summer of 2001, members of the Tate Geological Museum were excavating a large sauropod dinosaur (*Supersaurus vivianae*; see *Lovelace, Hartman & Wahl, 2007*) at the Jimbo Quarry in the Morrison Formation near Douglas, Wyoming (Fig. 1). WYDICE-DML-001 (aka the "Lori" specimen; see *Wahl, 2006*) was discovered during the removal of overburden from the quarry. The accidental nature of the discovery directly impacted the recovery of the delicately preserved specimen, resulting in some portions being damaged or lost during collection.

The Morrison Formation in central Wyoming is considered undivided adding to the difficulty in long distance correlation (*Trujillo, 2006*; *Trujillo, Chamberlain & Strickland, 2006*; *Trujillo et al., 2014*). Very fine (100 μm) euhedral zircons have been observed in heavy mineral separates from a smectite rich mudstone within 0.5 m above and a mixed smectite-illite mudstone below the interval in which WYDICE-DML-001 was discovered.

The Jimbo Quarry (Unit 1 of Fig. 2) is an isolated discrete unit with an uneven upper and lower surface that is interpreted to be a hyperconcentrated flow resulting from post-fire soil destabilization (*Lovelace, 2006*). WYDICE-DML-001 was discovered in a fine grained muddy sandstone that immediately overlies the Jimbo Quarry (Unit 2 of Fig. 2); this unit fines upward over 10–20 cm into the first 1.5 m of mixed smectite-illite mudstone of Unit 3 (Fig. 2). Unit 3 exhibits six discrete micritic limestone layers that overly the first 1.5 m of mudstone and are each differentiated by 20–50 cm of mixed smectite-illite mudstones with abundant charophytes and conchostraca (*Lovelace, 2006*).

The first four meters of strata overlying the Jimbo Quarry have been interpreted as representing a cyclical rise and fall of the local water table in a marginal lacustrine or wetland environment (*Lovelace, 2006*) similar to those seen in the Big Horn Basin of Wyoming (*Jennings, Lovelace & Driese, 2011*). The concentrated presence of barite in long bones and pleurocoels is consistent with a saturated microenvironment where free sulphur is available due to organic decay; this has been observed elsewhere in marginal lacustrine and wetland environments within the Morrison of Wyoming (*Jennings & Hasiotis, 2006*; *Jennings, Lovelace & Driese, 2011*). The interpretation of a wetlands or marginal lacustrine environment is supported by XRD of clay minerals, presence of

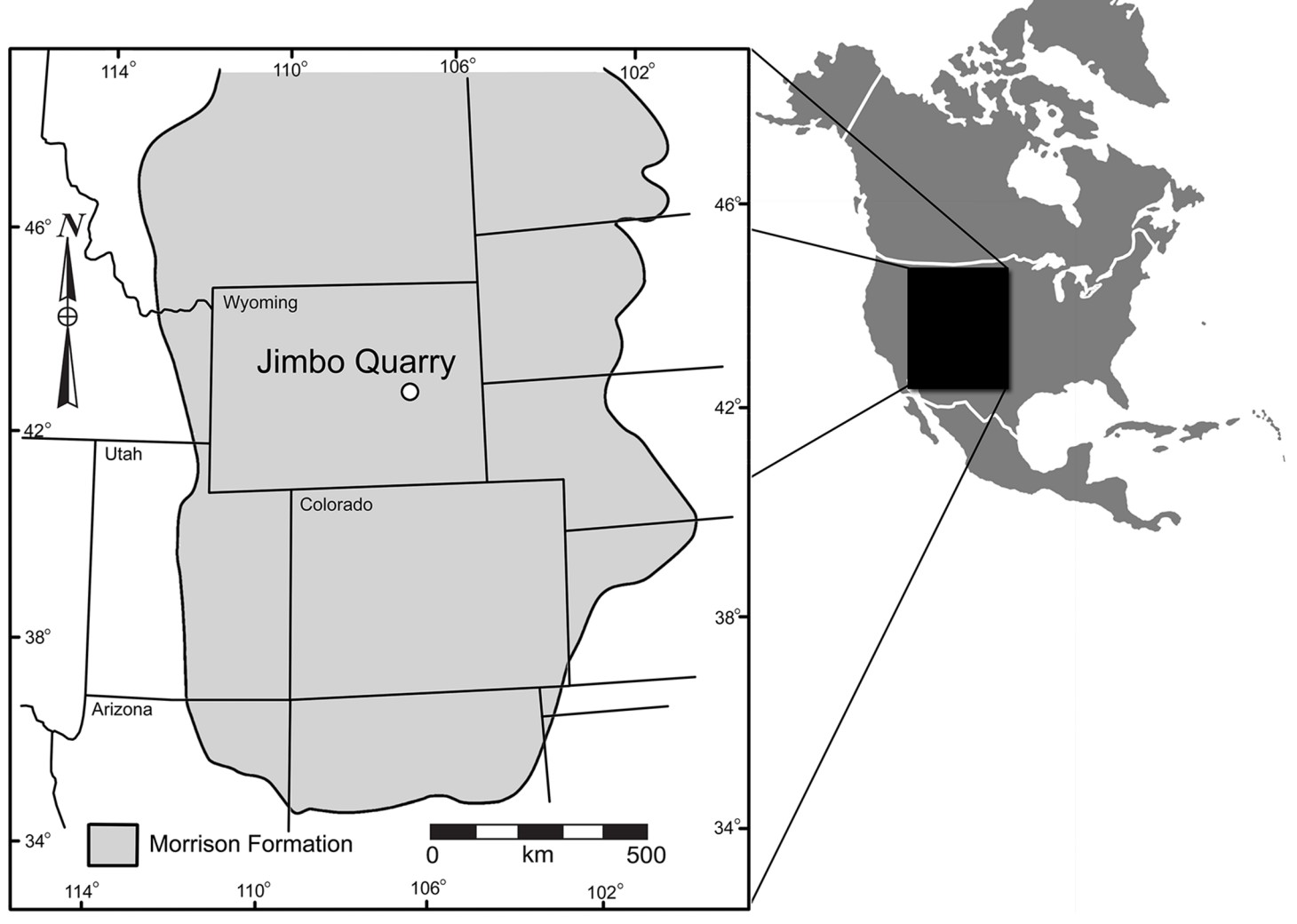

**Figure 1 Geographic relationship of the Jimbo Quarry and the majority of the Morrison Formation, Late Jurassic, USA.** Formation outcrop and map data based on paleobiodb.org.

freshwater algae and arthropods, and the lack of sedimentary structures indicative of fluvial transport.

WYDICE-DML-001 is preserved in partial articulation with little evidence of dissociation. The presence of organic material at the distal end of several manual and pedal unguals is consistent with the preservation of a keratinous sheath; no other soft-tissue preservation was observed. Much of the thoracic region is absent, although the relative positioning of the remaining elements (Fig. 3) suggests an animal in a resting position. Given the autochthonous nature of deposition it appears that *Hesperornithoides* was an inhabitant of wetland environments for at least a portion of its life history.

## DESCRIPTION

WYDICE-DML-001 has an estimated length of 89 cm (Figs. 4 and 5; Table 1). The hind legs are folded in a crouching or resting position, the head is turned to the side underneath

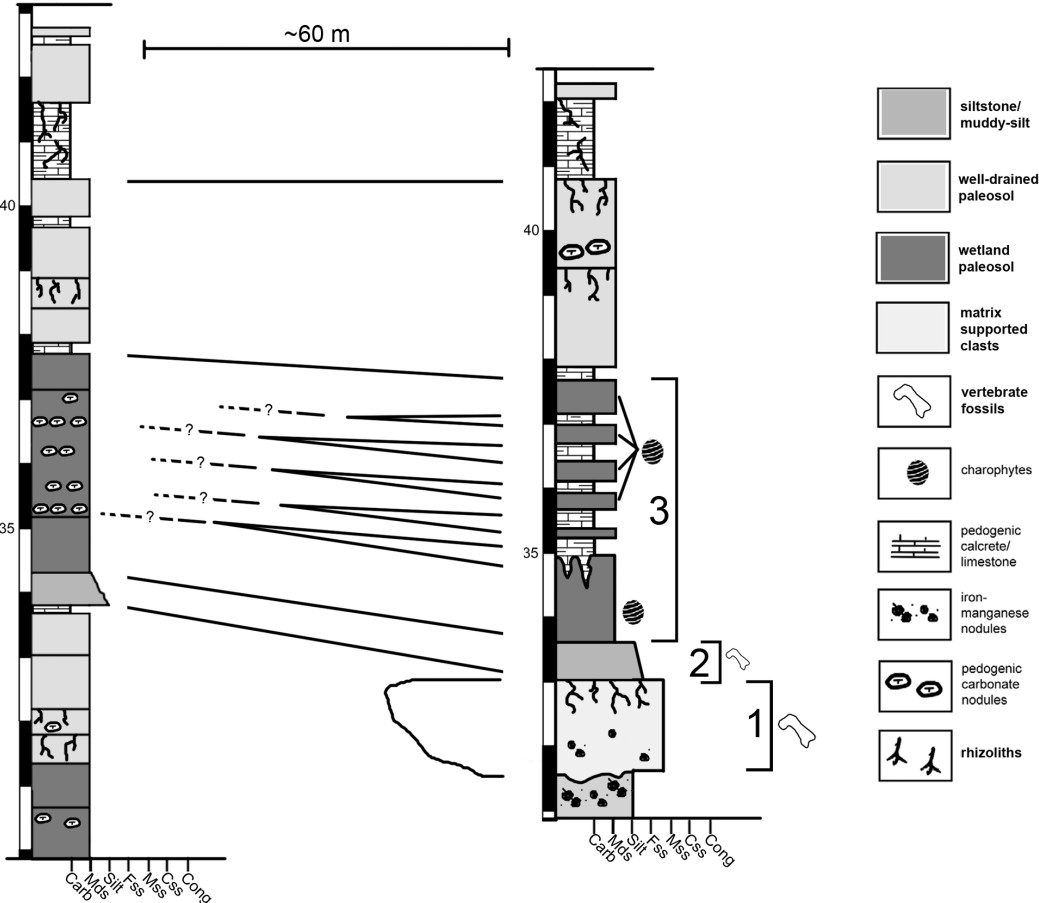

**Figure 2 Condensed stratigraphic sections demonstrating the lateral variability near the Jimbo Quarry.** "?" indicate loss of direct lateral correlation due to covered section. 1 = Jimbo Quarry; 2 = Lori locality; 3 = marginal wetland deposits.

the left manus, and the preserved mid-caudal series wraps around the torso, reminiscent of the sleeping posture preserved in *Mei* and *Sinornithoides* (*Xu & Norell, 2004*; *Gao et al., 2012*; *Russell & Dong, 1993*). *Hesperornithoides* is compared below both to other paravians and to other small Morrison coelurosaurs.

## Ontogenetic status

Visual inspection under 100× magnification showed neural arches are fused with fully obliterated synchondroses sutures on all preserved vertebrae, ruling out a hatchling or juvenile individual (sensu *Hone, Farke & Wedel, 2016*). Adult or subadult status is reinforced by general skeletal proportions (Fig. 5; Tables 1 and 2), as WYDICE-DML-001 lacks a relatively enlarged cranium or other allometric proportions associated with early ontogenetic stages. WYDICE-DML-001 does not exhibit signs of advanced ageing, as the skull and proximal tarsal sutures lack evidence of obliteration by co-ossification. Without histological analysis the ontogenetic stage cannot be resolved beyond "adult or subadult," but either designation would result in minimal additional linear skeletal growth, making the estimated one meter total length (Table 1) of *Hesperornithoides* substantially

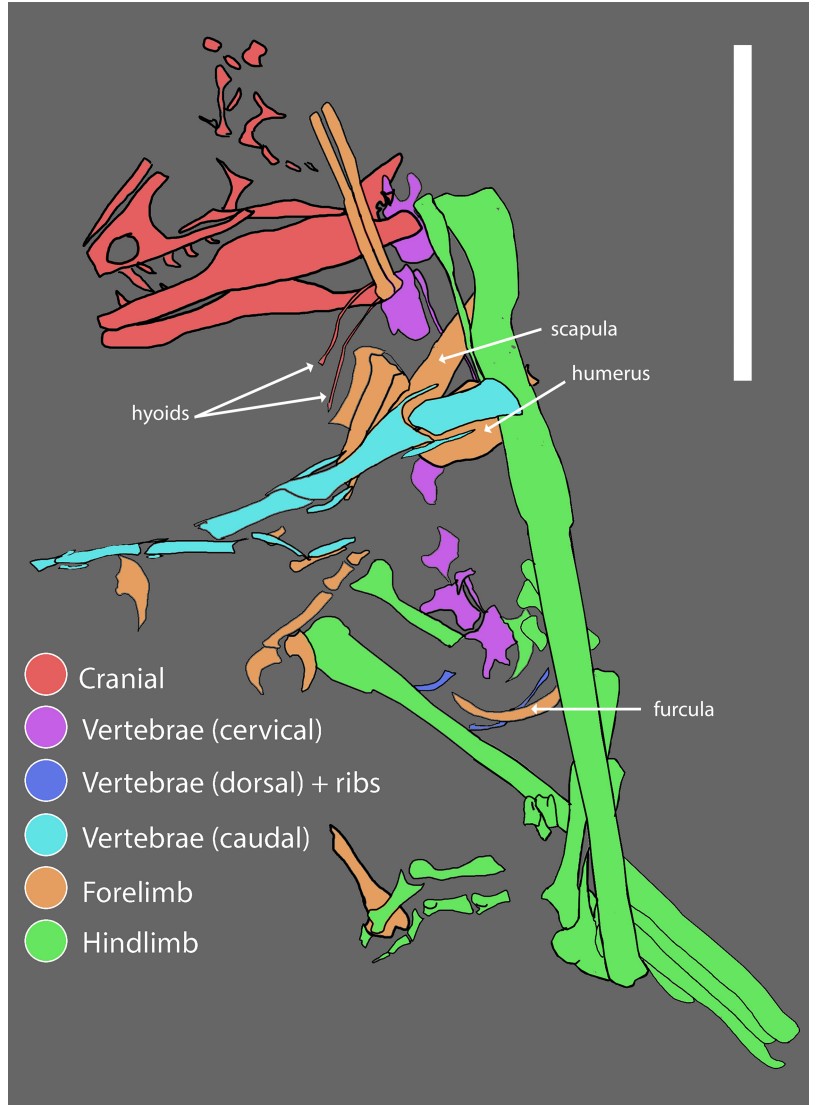

**Figure 3 Reconstructed quarry map of WYDICE-DML-001.** Association of skeletal elements assembled from 3D scans of specimen blocks prior to final mechanical preparation. Scale bar = 6 cm.

smaller than other relatively complete theropods from the Morrison Formation (*Foster, 2003*). The same can be said about individual elements, for example the humerus of WYDICE-DML-001 is 29% the length of the *Coelurus* holotype, 17% the length of the *Tanycolagreus* holotype, and 28% the length of the *Ornitholestes* holotype.

## Skull

Cranial elements are preserved in a separate "skull block" (Figs. 3 and 4) prepared so both the right and left skull elements are largely visible in lateral view; some palatal elements are visible on the right side of the block. The right jugal, lacrimal, and posterior process of the maxilla are articulated and well preserved; the right quadrate is also exposed (Fig. 6). The skull and mandible exhibit some lateral compression with the left side being dorsally

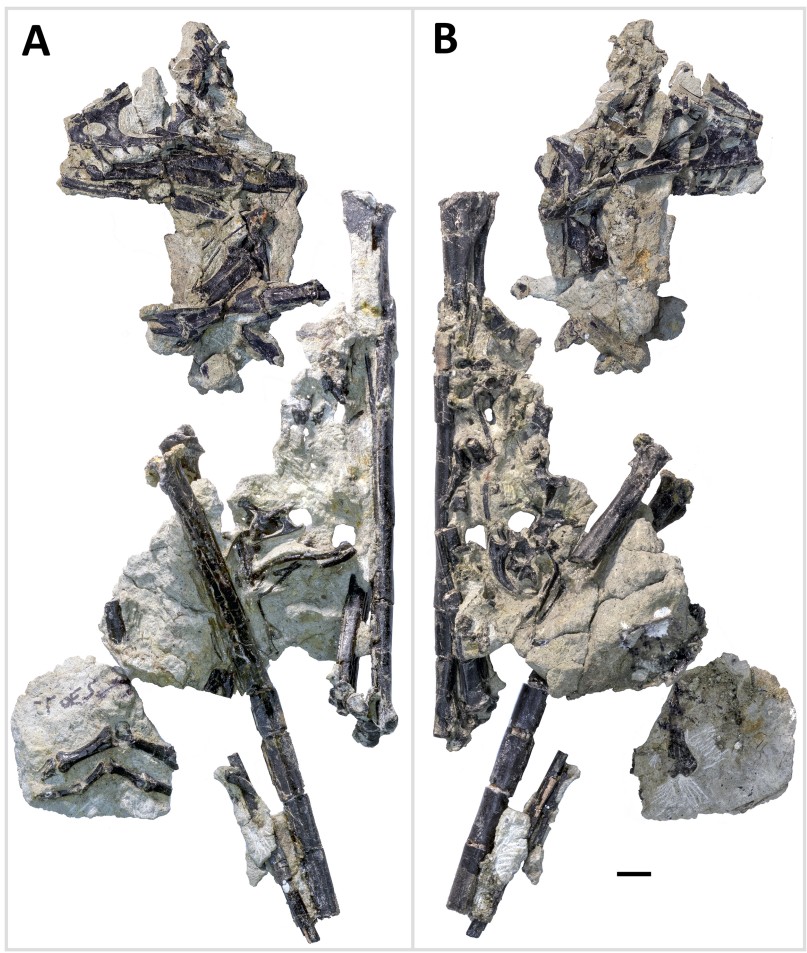

**Figure 4** **Primary blocks of WYDICE-DML-001.** "Left" (A) and "right" (B) sides of the blocks after final preparation (B). Scale bar = one cm. Images taken by Levi Shinkle, used with permission.

displaced (Fig. 6). The braincase suffers from both inadvertent damage during collection (some natural molds of broken elements exist) and pre-depositional disruption of elements. Posterior skull roof elements are progressively displaced dorsally; the left jugal is dorsoventrally rotated out of position underlying the posterior process of the maxilla. A portion of the lower half of the left lacrimal is exposed in lateral view; a medial impression of the mid-lacrimal is also visible. The premaxillae, as well as anteriormost portions of the maxillae, nasals, and dentaries were unfortunately destroyed during discovery of the specimen.

The skull is triangular in lateral aspect as seen in most basal paravians and enough of the snout is preserved to show the external naris was not enlarged. The maxilla exhibits an extensive, sharp-rimmed antorbital fossa containing a D-shaped antorbital fenestra, a large maxillary fenestra, and the posterior margin of a promaxillary fenestra (Fig. 6). This rim is also found in derived members of Sinovenatorinae (*Xu et al., 2002*: Fig. 1a; *Xu et al., 2011*: Fig. S1a; *Xu et al., 2017*: Fig. 2a), Microraptoria (*Xu & Wu, 2001*: Fig. 4B; *Pei et al., 2014*: Fig. 2), and some archaeopterygid specimens (*Pei et al., 2017a*: Fig. 5;

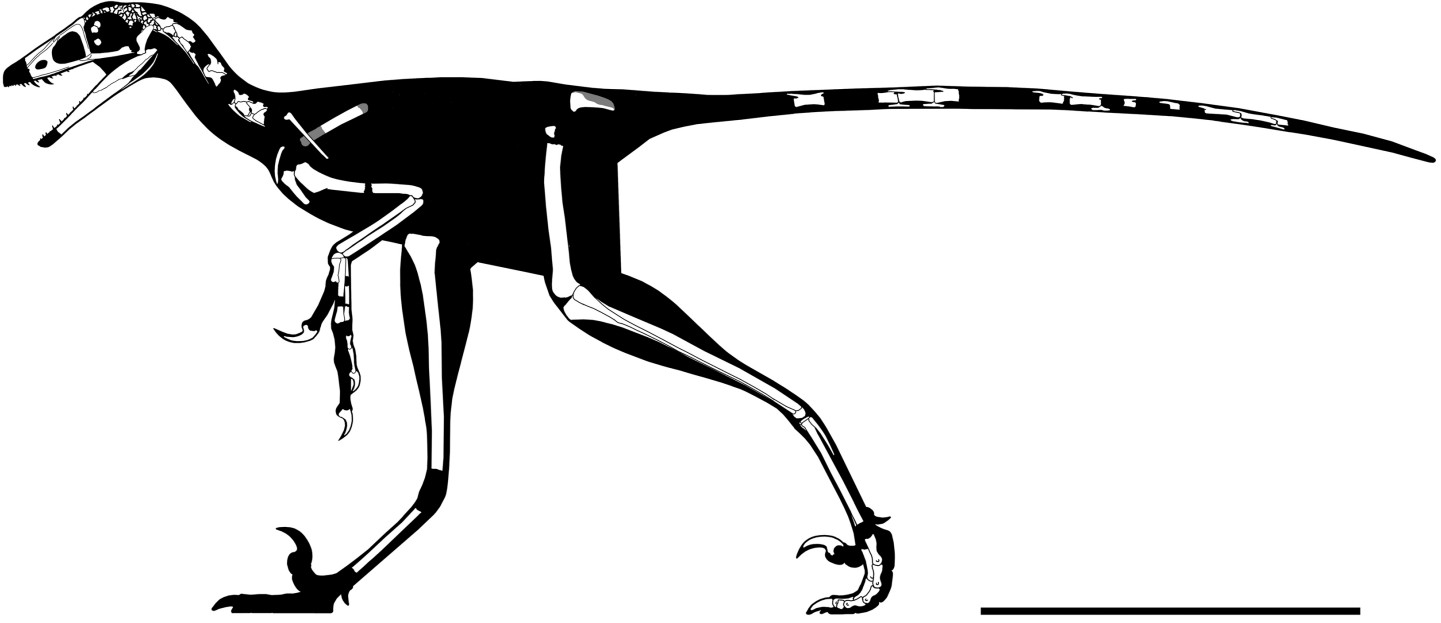

**Figure 5** **Rigorous skeletal reconstruction of WYDICE-DML-001.** Scale bar = 25 cm.

*Rauhut, Foth & Tischlinger, 2018*: Fig. 8A). Similar to most dromaeosaurids and *Halszkaraptor* (*Godefroit et al., 2008*: Fig. 4; *Lu & Brusatte, 2015*: Fig. 2; *Zheng et al., 2009*: Fig. 2a; *Cau et al., 2017*: Fig. 3b), the maxillary fenestra is dorsally displaced within the antorbital fossa (Fig. 7). The maxillary fenestra is positioned far from the unpreserved anterior edge of the antorbital fossa and is not set within a concavity unlike some dromaeosaurids. There is an accessory fossa posteroventral to the fenestra, however, as in *Zhenyuanlong* (*Lu & Brusatte, 2015*: Fig. 2) but less developed than in microraptorians. A robust external ascending process separates the nasal and elongate antorbital fossa until the anterior margin of the antorbital fenestra. Segmented CT data shows medially there is an extensive palatal shelf placed more ventrally than in eudromaeosaurs (Fig. 7), and posteriorly the jugal process is shallow. Both promaxillary and epiantral recesses are developed, the postantral pila does not extend into the antorbital fenestra as it does in some dromaeosaurids, and the palatal shelf does not extend dorsally to be visible laterally as it does in *Saurornitholestes* and *Atrociraptor* (Fig. 7). Compared to *Ornitholestes*, the maxilla has a larger maxillary fenestra, a medial fenestra for the maxillary antrum (*Witmer, 1997*:42) and an external dorsal process that ends sooner so that the nasal contacts the antorbital fossa.

The incomplete nasals are unfused with smooth external surfaces which have no significant transverse convexity. The portion adjacent to the antorbital fossa on the left nasal lacks accessory pneumatic foramina. Both lacrimals are preserved, lacking horns but possessing the lateral expansion on the dorsal edge typical of pennaraptorans. A foramen is present in the posterodorsal corner of the antorbital fossa, here scored as pneumatic, but given similar structures have been considered to house the lacrimal duct (e.g., Fig. 5A in *Yin, Pei & Zhou (2018)* for *Sinovenator*), more objective criteria are needed. The

Table 1 Measurements of the axial skeleton of WYDICE-DML-001.

| Element | Longest distance (mm) | Additional measurement (mm) |
| --- | --- | --- |
| Skull | | |
| Ventral length* (maxilla to quadrate) | 39.5 | |
| Longest maxillary tooth crown | 6.4 | |
| Shortest erupted tooth crown | 2.1 | |
| Mandible* (missing anterior end) | 42.5 | |
| Axial (centra length) | | |
| CV3 | 11.7 | |
| CV4 | est 14 | |
| CV?6 | 14.5 | |
| CV8 | 13.7 | |
| CV9/D1 | 11.7 | |
| Proximo/mid caudal | 20.8 | |
| CA B (mid caudal) | 22.2 | max height = 6 |
| CA C (mid caudal) | 22.3 | |
| CA E (mid caudal)* | 23.6 | max height = 4.9 |
| CA H (distal caudal) | 18.2 | 3.2 |
| CA I (distal caudal) | 17.5 | 2.5 |
| Body length estimates | | |
| Head | 66 | |
| Neck | 108 | |
| Dorsals | 130 | |
| Sacrum | 45 | |
| Caudals | 540 | |
| Total | 889 | |

**Note:**
  * Indicates incomplete or partially restored elements.

posterior lacrimal process is short (Fig. 8), unlike most long-tailed paravians, but also present in *Zanabazar*, *Archaeopteryx*, and *Epidexipteryx* (*Barsbold, 1974*: plate 1 Fig. 1b; *Rauhut, 2013*: Fig. 1A; *Zhang et al., 2008*: Fig. 1c). The lacrimal differs from *Tanycolagreus* in lacking a dorsally projecting horn (*Carpenter, Miles & Cloward, 2005a*: Fig. 2.4G), and from *Ornitholestes* in having a laterally projecting antorbital process, both more similar to maniraptoriforms. The ventral process is strongly expanded distally and has a lateral lamina which never fully overlaps the medial lamina. The jugal is dorsoventrally low but lateromedially compressed beneath the orbit and laterotemporal fenestra. Segmented CT data indicates that unlike most maniraptorans (exceptions are *Zanabazar*, *Velociraptor*, and *Deinonychus*; *Norell et al., 2009*:34; *Barsbold & Osmólska, 1999*:200; *Witmer, 1997*:45) and *Ornitholestes* (*Witmer, 1997*:45), the jugal is pneumatic (Fig. 9) with a large opening in the antorbital fossa. The anterior end is only slightly expanded and there is no foramen medially at the level of the postorbital process. Along its ventral edge, there is a longitudinal ridge extending most of the way under the orbit. The jugal and postorbital contact completely separates the orbit from the infratemporal fenestra.

**Table 2 Measurements of the appendicular skeleton of WYDICE-DML-001.**

| Appendicular element | Longest distance (mm) | Other measurement (mm) |
| --- | --- | --- |
| Coracoid* | Height 21.4 | Width 14.6 |
| Left humerus* (proximal portion) | 34.5 | Shaft 32.5 |
| Left ulna | 46.4 | |
| MCI | 16.9 | |
| MCII + carpals* | 61 | |
| Manus phalanx III-2 | 7 | |
| Manus phalanx III-3 | 18.2 | |
| Manual ungual I | 17.1 | |
| Manual ungual II | 15.5 | |
| Manual ungual III | 14.4 | |
| Left femur* (no proximal end) | 93.9 | |
| Left tibia | 168 | |
| Right tibia* (missing distal end) | 153 | |
| Left MT packet* (proximal) | 55.2 | |
| Left pes phalanx III-1 | 26.3 | |
| Left pes phalanx III-2 | 17.3 | |
| Left pes phalanx IV-2 | 13.2 | |
| Left pes phalanx IV-3* | est 8.5 | |
| Left pes phalanx IV-4 | 6.9 | |
| Pes ungual III | 13.6 | Height 3.7 |

**Note:**
* Indicates incomplete or partially restored elements.

The postorbital, squamosal, quadratojugal, palatine, and perhaps pterygoid cannot be exposed via mechanical preparation in sufficient detail to score characters from. Interference from barite inclusions in the anterior cervical vertebrae have frustrated multiple CT scan attempts. Proportions of suborbital bones suggest a parietal shorter than the frontal. Both quadrates are preserved and partially exposed from surrounding matrix, showing a planar articulation with the quadratojugal and bicondylar distal articulations. CT scans show the quadrate is pneumatized as in troodontids but unlike *Ornitholestes*, with a fossa on the posterior surface. The quadrate partially encloses the small paraquadrate foramen laterally unlike *Tanycolagreus* (*Carpenter, Miles & Cloward, 2005a*: Fig. 2.4O) and other maniraptorans (Fig. 10; compare to, e.g., *Barsbold, Osmolska & Kurzanov, 1987*: plate 49 Fig. 4; *Xu & Wu, 2001*: Fig. 4D; *Burnham, 2004*: Fig. 3.10B; *Hu et al., 2009*: Fig. S2d; *Gao et al., 2012*: Fig. 2A; *Xing et al., 2013*: Fig. S1; *Cau et al., 2017*: Fig. 3a; *Gianechini, Makovicky & Apesteguía, 2017*: Fig. 5; *Yin, Pei & Zhou, 2018*: Fig. 7A). Note in Fig. 10 there is a vertical suture at the top center of the closeup image and that bone to the left of that is a fragment of quadratojugal adhering to the quadrate.

## Mandible and dentition

The left and right mandibles are complete save for the anterior portion of the dentaries that were destroyed during discovery. The lateral surface of the posterior left mandible is well

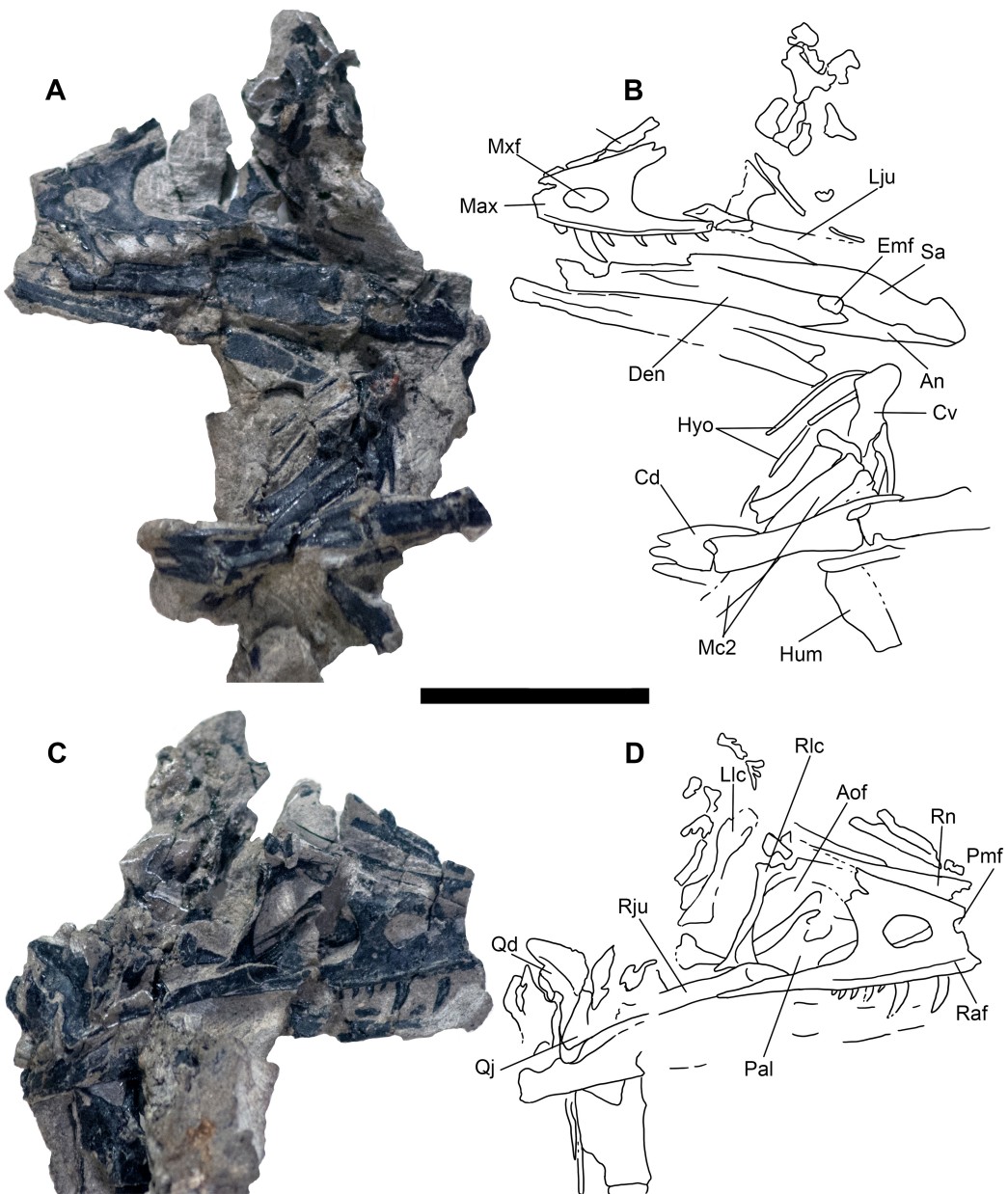

**Figure 6 Skull block and interpretive drawing.** Skull block in left lateral (A and B) and right lateral (C and D) views. Abbreviations: An, angular; Aof, antorbital fenestra; Cd, mid caudal vertebrae; Cv, cervical vertebra; Den, dentary; Emf, external mandibular fenestra; Hyo, hyoids; Hum, humerus; Lju, left jugal; Llc, left lacrimal; Max, maxilla; Mc2, metacarpal II; Mxf, maxillary fenestra; Pal, palatine; Pmf, promaxillary fenestra; Qd, quadrate; Qj, quadratojugal; Raf, ridge under antorbital fossa; Rju, right jugal; Rlc, right lacrimal; Rn, right nasal; Sa, surangular. Scale bar = five cm. Photo credit Levi Shinkle, used with permission.

exposed, however, the dorsolateral dentary is largely obscured by the overlying maxillary teeth and associated matrix and must be described from segmented CT scan data. The posteriormost right mandible is laterally exposed and the medial surface of the dentary is visible on the left side of the skull block. The labial dentary groove widening posteriorly is clearly visible; initially described as a troodontid character (*Hartman, Lovelace & Wahl, 2005*;

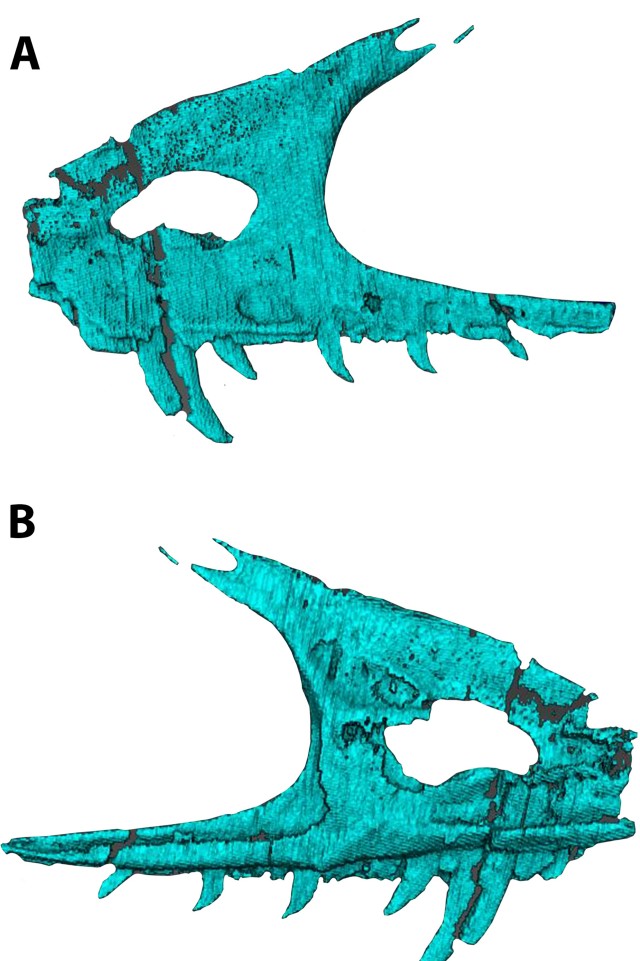

A

B

**Figure 7 Segmented left maxilla of WYDICE-DML-001.** Shown in lateral (A) and medial (B) views.

*Wahl, 2006*) it is here resolved as a symplesiomorphy seen also in some microraptorians, unenlagiines, *Halszkaraptor*, and most archaeopterygids (*Xu & Wu, 2001*: Fig. 6; *Paul, 2002*: plate 7A; *Gianechini & Apesteguia, 2011*: Fig. 2Ag; *Pei et al., 2014*: Fig. 2; *Cau et al., 2017*:S23; *Gianechini, Makovicky & Apesteguĺa, 2017*: Fig. 1-2; *Lefèvre et al., 2017*: Fig. 2; *Pei et al., 2017a*: Fig. 5). The groove is absent in *Ornitholestes*, however, and the dentary differs from that referred to *Coelurus* in not being downturned (*Carpenter et al., 2005b*: Fig. 3.3). Whether the new taxon's dentaries are entirely straight or slightly upturned anteriorly is uncertain due to the missing anterior tips. The slender dentaries lack a posterolateral shelf and have a deep Meckelian groove positioned at approximately midheight. *Hesperornithoides* has a small external mandibular fenestra (<11% of mandibular length; Fig. 5B) unlike most other maniraptorans except for *Zhenyuanlong* and *Dromaeosaurus* (*Currie, 1995*: Fig. 7A; *Lu & Brusatte, 2015*: Fig. 2). No coronoid is obvious despite a well preserved in situ mandible, but we conservatively score the taxon unknown given the reduced state of this element in some maniraptoriforms. Posteriorly, the surangular is shallower than the dentary at the anterior border of the

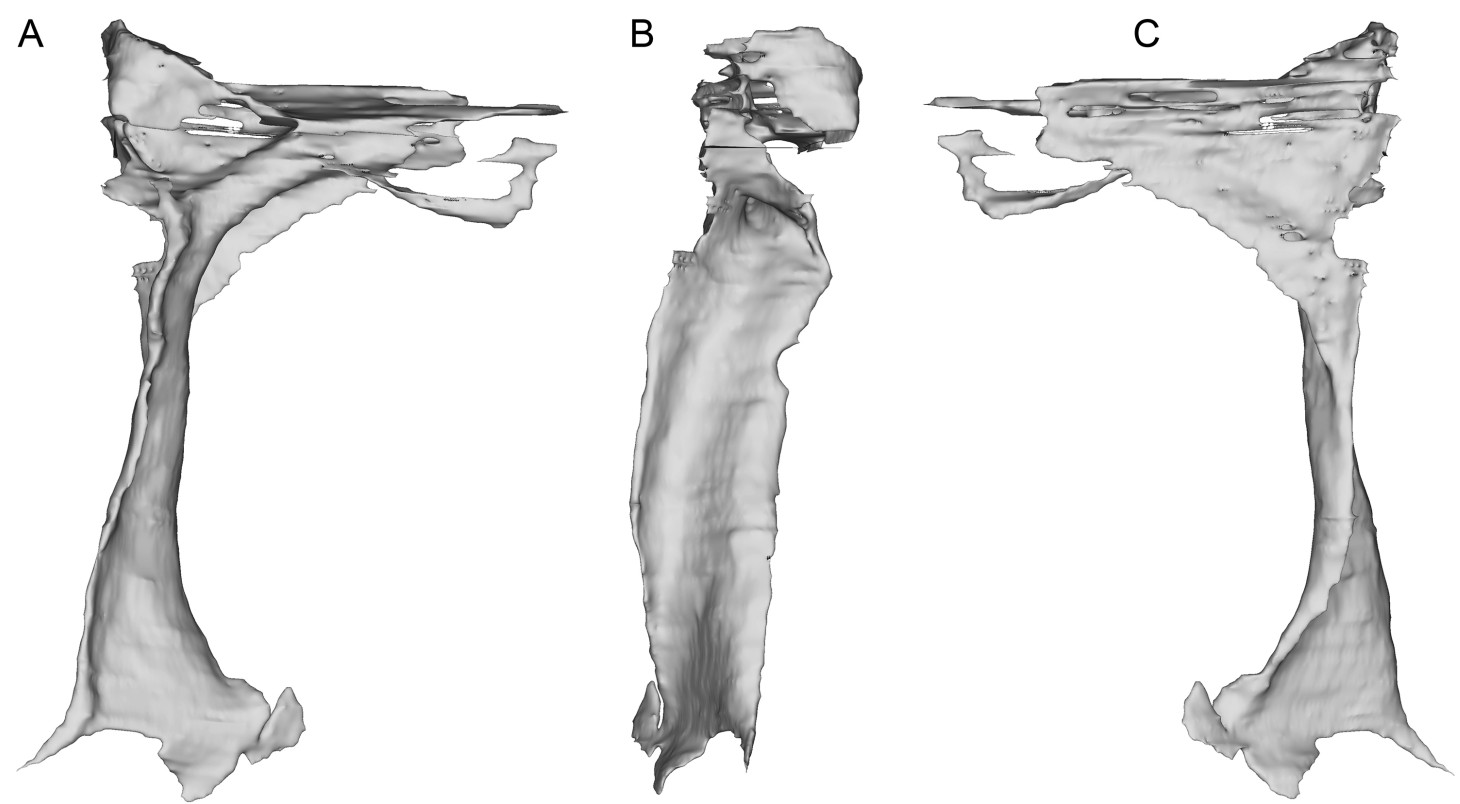

**Figure 8 Segmented right lacrimal of WYDICE-DML-001.** Shown in lateral (A), posterior (B), and medial (C) views.

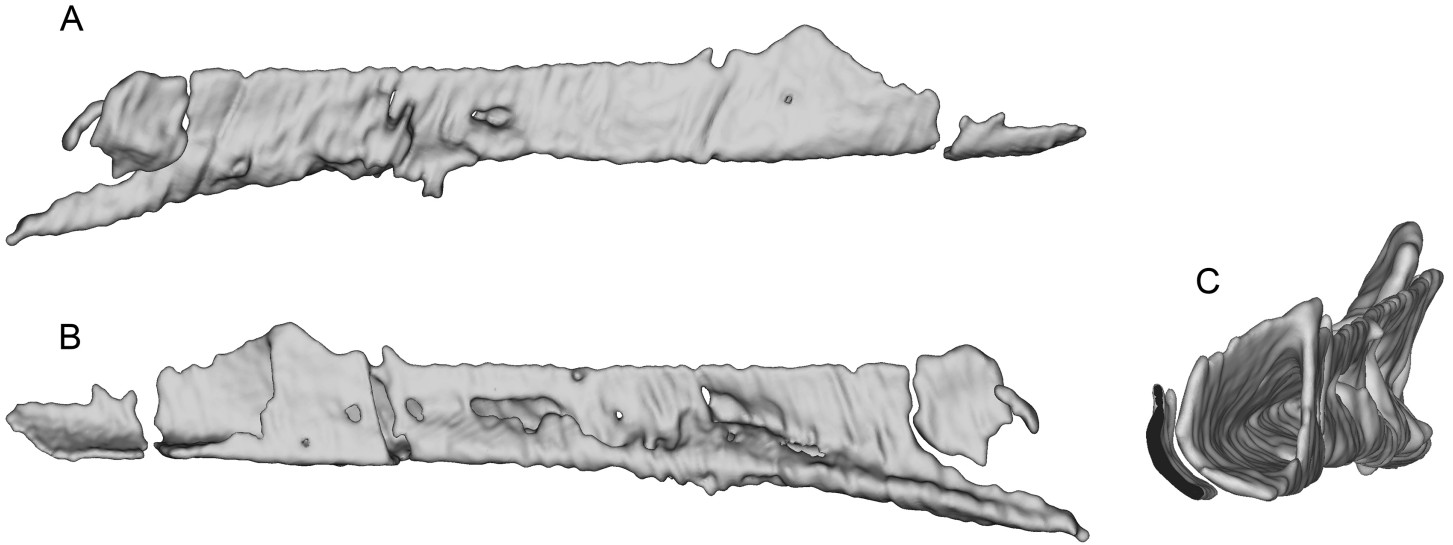

**Figure 9 Segmented left jugal of WYDICE-DML-001.** Shown in medial (A), lateral (B), and anterolateral oblique (C) views.

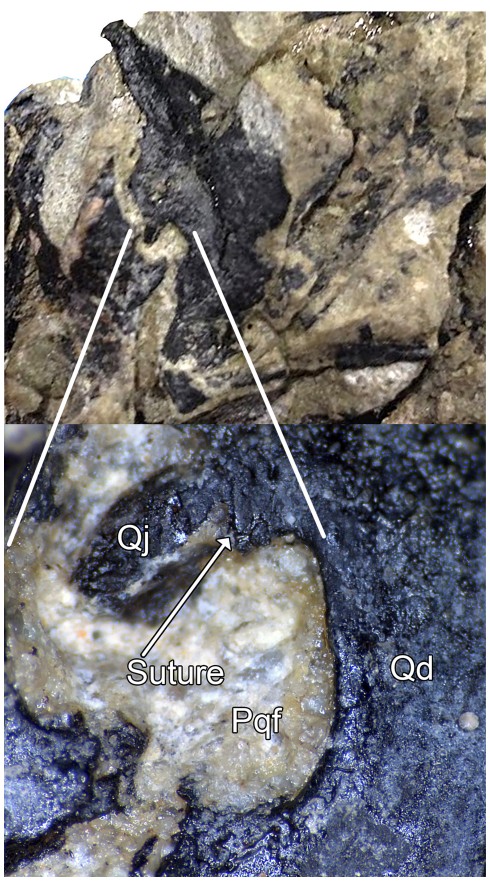

**Figure 10 Paraquadrate foramen of WYDICE-DML-001.** Paraquadrate foramen and inset detail showing the contribution of the quadrate to the paraquadrate foramen. Abbreviations: Pqf, paraquadrate foramen; Qd, quadrate; Qj, quadratojugal.

external mandibular fenestra (Fig. 6A), has a very long suture with the dentary anterior to this fenestra, and lacks a pronounced coronoid eminence and or a process invading the fenestra.

The lack of anteriormost portions of the mandible and maxilla prevent determination of changes in size or spacing of alveoli, but in both elements the posterior teeth are smallest and the left maxilla preserves an anterior crown base smaller than the next tooth, the largest in the series. At least 10 maxillary teeth and 11 dentary teeth were present, with the total count being perhaps ~14 and ~17. Teeth are relatively large and recurved and possess well developed mesial serrations as in most dromaeosaurids, *Troodon* and *Caihong* (*Hu et al., 2018*:4). The serrations on the mesial carinae of maxillary teeth are smaller than the distal serrations as in basal dromaeosaurids. Mesial serrations are restricted to the apical third of the crown and appear absent in some teeth. Serrations are small (5.5 per mm distally) as in *Sinusonasus*, *Liaoningvenator* and some sinovenatorines, and not apically hooked. Similar to the condition described for *Serikornis*, the maxillary teeth are anisodont with crown height of the largest exposed teeth twice the size of others (*Lefèvre et al., 2017*: Fig. 2). There is a slight mesiodistal constriction between tooth root and crown in fully erupted teeth (Fig. 11G) unlike *Ornitholestes* and most dromaeosaurids except some

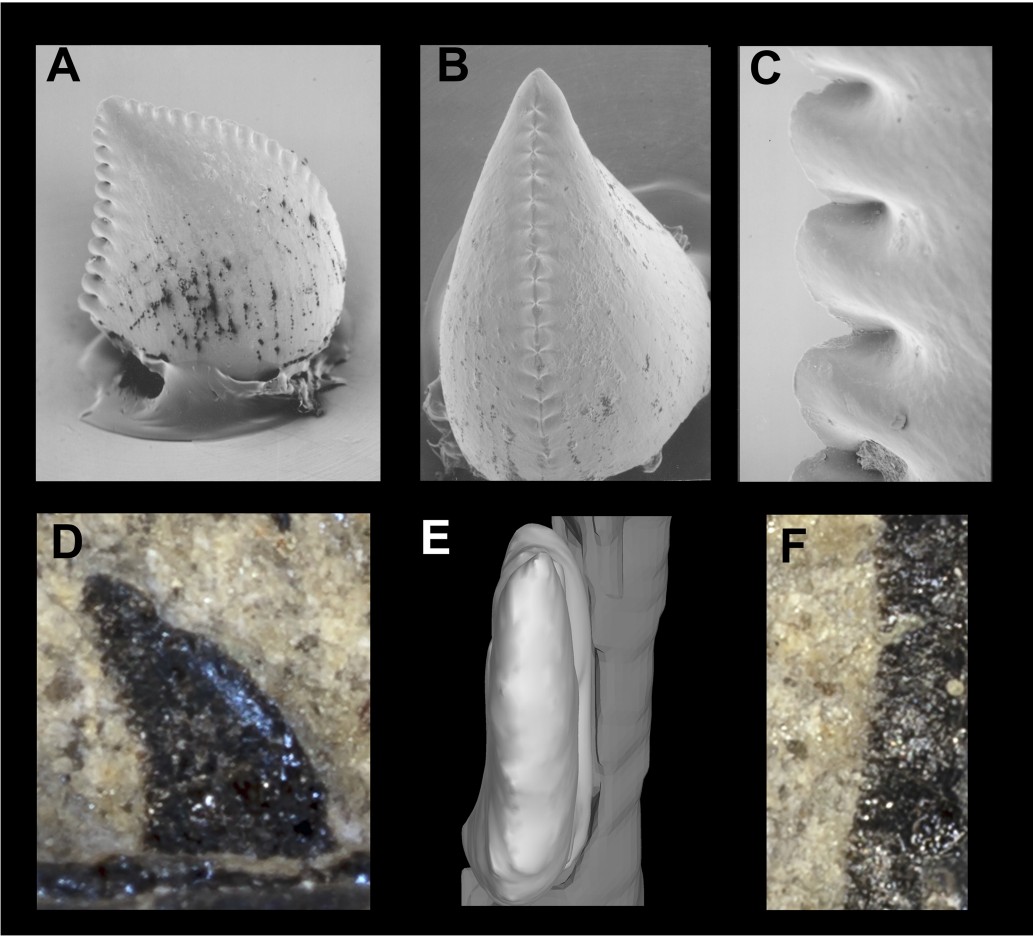

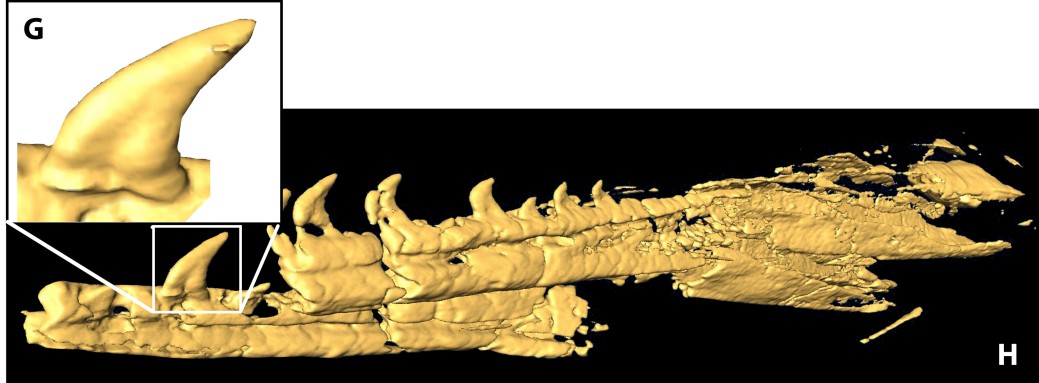

**Figure 11 Mandible and comparative tooth morphology of *Koparion* and *Hesperornithoides*.** Tooth of *Koparion* in "side" (A) and mesial (B) views, and a detail of the distal serrations (C). *Hesperornithoides* maxillary tooth in labial view (D), rendered from CT data in mesial view (E) and a detail of the distal serrations (F). Segmented CT data of fully erupted tooth (G) and anterior mandibular elements of *Hesperornithoides* in oblique view (H). Photo credit (A–C): Dan Chure.

microraptorians (*Xu, Zhou & Wang, 2000*: Fig. 2d; *Xu & Li, 2016*: Fig. 3), although no proximal expansion of the root is present. Both root and crown are labiolingually compressed, and the enamel shows no trace of longitudinal grooves. *Chure (1994)*

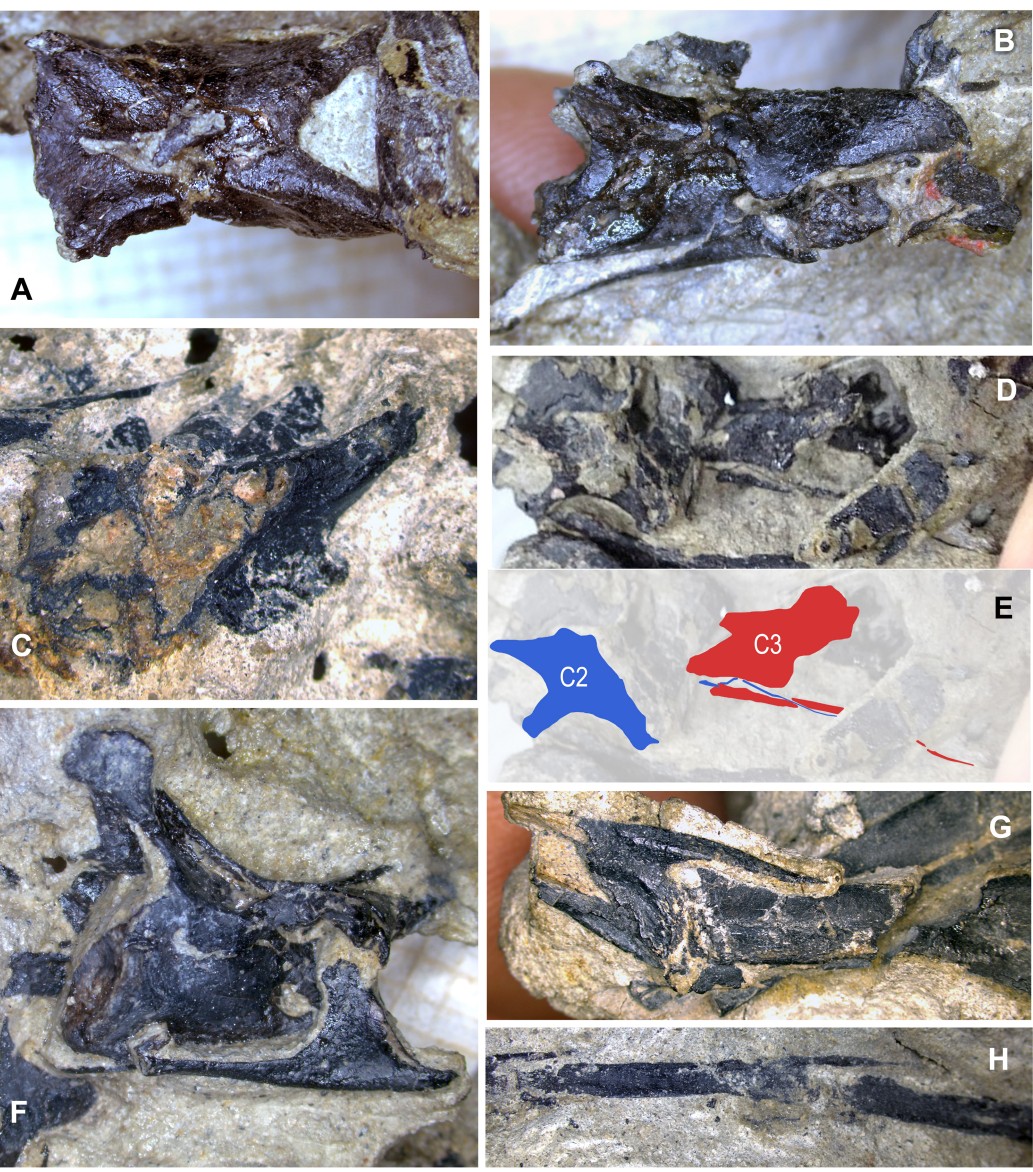

**Figure 12 Select axial elements of WYDICE-DML-001.** Cervical vertebra three in dorsal (A) and right lateral (B) views. (C) Mid-cervical vertebra in cross-section. Photo and schematic of association between axis and cervical three (D and E). (F) Cervicodorsal in right lateral view. Articulated middle (G) and distal (H) caudal vertebrae in right lateral views.

previously described an isolated tooth from the Morrison Formation of Utah as the troodontid *Koparion douglassi*. This tooth differs from *Hesperornithoides* teeth in being more recurved, labiolingually wide (Basal Width/FABL ~0.72 compared to ~0.45), possessing large serrations as in derived troodontids, exhibiting mesial serrations that extend to within two serration lengths of the crown base, and possessing blood pits (Fig. 11). While it may belong to a derived troodontid, serration size and extent was extremely homoplasic in our phylogenetic results suggesting caution should be applied. Paired hyoids are preserved as thin parallel rod-like structures on the left side
of the skull block; posteriorly the hyoids overlie the medioposterior surface of the right mandible.

## Axial skeleton

The axial skeleton is distributed across three blocks (Fig. 4). Six presacral vertebrae are preserved on the "body block," three anterior cervicals, two mid-cervicals and a cervicodorsal (Fig. 12). Three articulated mid-anterior caudals are found on the skull block, and the remaining caudal vertebrae are preserved on the "hand block," some as natural molds. Cervical centrum measurements indicate a cervicofemoral ratio (~0.95) similar to anchiornithines (0.81–1.01), troodontids (0.83–1.08), and dromaeosaurids (1.02–1.04) but shorter than *Halszkaraptor* (2.65) or *Archaeopteryx* (1.21–1.37). The axis is partially preserved, possessing epipophyses that extend past the postzygapophyseal tips. The third cervical centrum is over four times longer than tall, posteriorly extending past the neural arch, with amphicoelous articular surfaces, a transversely convex ventral surface and a single pair of pleurocoels posterior to the parapophysis (Fig. 12B). CT cross sections and broken surfaces reveal additional cavities, including those extending into the zygapophyses (Fig. 12C). The neural spines are long and low, centered on the neural arch. Contra initial reports (*Hartman, Lovelace & Wahl, 2005*; *Wahl, 2006*) some of the cervical ribs extend beyond the posterior margin of the centra. Cervical ribs begin broad in cross-section but thin rapidly in width to an almost hair-like diameter (Fig. 12E). Initial mechanical preparation missed this thinning, and it appears that other paravians described as having short cervical ribs may also be in need of additional preparation to ensure accurate scoring of this character (S. Hartman, 2016, personal observation). Two cervical vertebrae are preserved with fused cervical ribs, consistent with a subadult or adult individual.

A single anterior dorsal vertebra is preserved (Fig. 12F). Its centrum is longer than wide or tall, posteriorly concave and has a pleurocoel behind the parapophysis with no lateral fossa. The anteroventral centrum is hidden by matrix, preventing determination of hypapophyseal height. Separated hyposphenes are present posteriorly. The transverse processes are short according to our newly quantified version of this classic TWiG character. It has a fan-shaped and moderately tall neural spine.

Portions of at least three caudals are preserved on the hand and skull blocks, an additional nine caudals are preserved on the hand block, several as natural molds. The 12 caudals provide good representation from mid and distal portions of the tail (Figs. 5, 8G and 8H). Caudal morphology suggests a distinct transition from shorter proximal caudals to elongate mid and distal caudal centra. The neural spines transition from well-developed on more proximal caudals to absent in mid caudals, whereas *Coelurus* has low spines even on distal caudal vertebrae (*Makovicky, 1995*). The distal caudals have transversely flat dorsal surfaces between the zygapophyses, and while the distalmost elements develop a negligible concavity, it is never comparable to the sulcus found in other troodontids. Distal caudal prezygapophyses are between 33% and 100% of central length (Figs. 12H and 13), as in *Scansoriopteryx*, some troodontids, *Caihong* and jeholornithids (*Currie & Zhiming, 2001*: Fig. 5A; *Czerkas & Yuan, 2002*: Fig. 13; *Zhou & Zhang, 2002*: Fig. 1b-c;

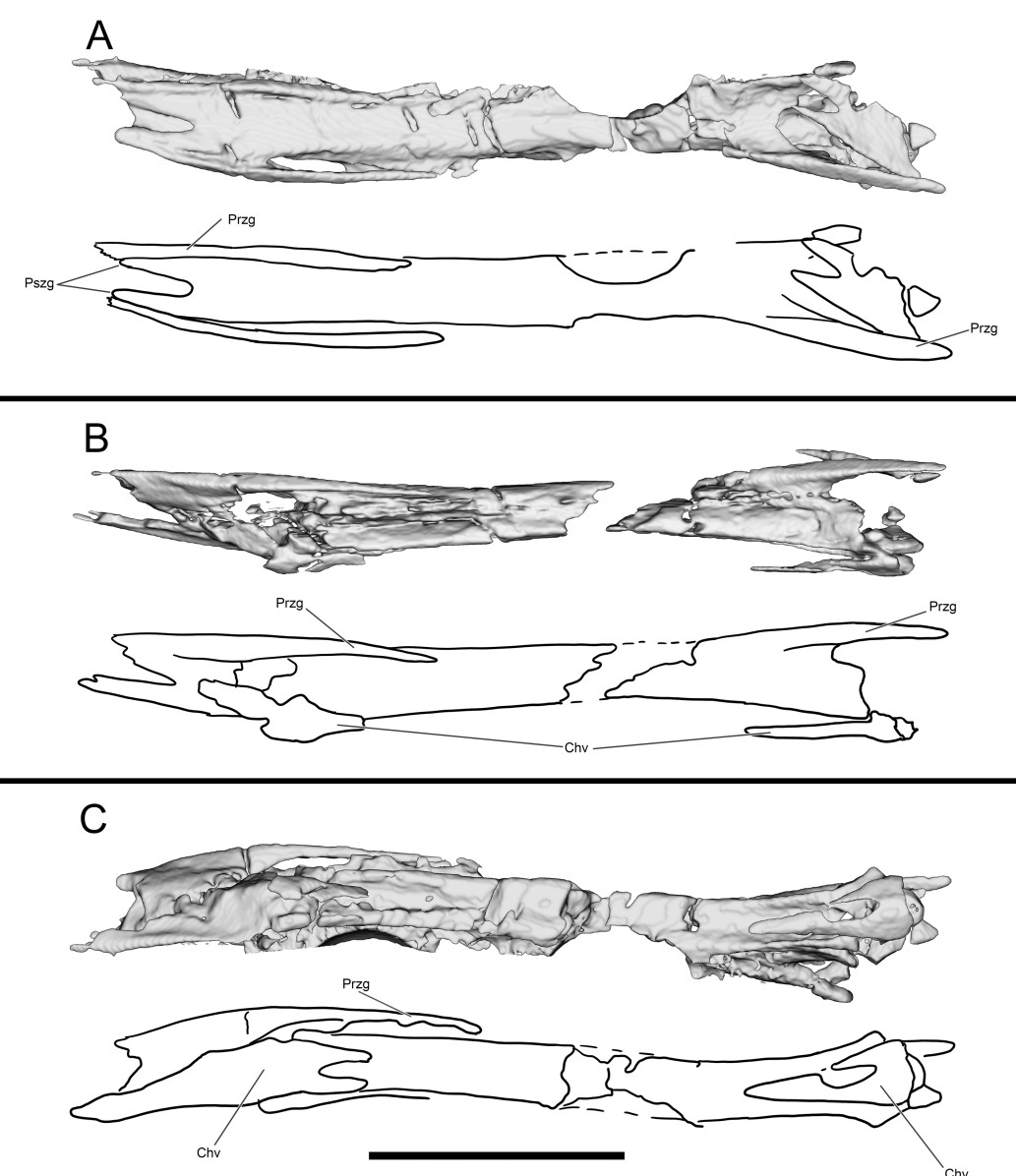

**Figure 13 Segmented and interpretive drawing of distal caudal vertebrae of WYDICE-DML-001.** In dorsal (A), right lateral (B) and ventral (C) view. Anterior is to the right in all views. Abbreviations: Chv, chevron; Przg, prezygapophysis; Pszg, postzygapophysis. Scale bar = one cm.

*Xu & Wang, 2004*: Fig. 1; *Norell et al., 2009*: Fig. 32A-B; *Hu et al., 2018*: Fig. 2e) but unlike the shorter prezygapophyses of *Coelurus* (*Carpenter et al., 2005b*: Fig. 3.6C). Unlike microraptorians and eudromaeosaurs, these processes lack bifurcation. The mid and distal caudals have a longitudinal sulcus on the lateral surface where centra meet neural arches, which is primitive for paravians, being present in unenlagiines, *Liaoningvenator*, archaeopterygids, and *Jeholornis* (*Motta, Egli & Novas, 2017*:174; *Shen et al., 2017a*: Fig. 4A). CT scans reveal pneumatopores within a fossa on the lateral surface of some caudal vertebrae. Ventrally, distal caudal centra exhibit a deep longitudinal groove. Mid caudal

chevrons are dorsoventrally flat but without highly elongate processes, and bifid anteriorly and posteriorly unlike those of *Ornitholestes* (*Carpenter et al., 2005b*: Fig. 3.6B) or other non-paravians.

## Pectoral girdle and forelimb

The pectoral girdle is poorly preserved, but a large and robust furcula was preserved in association with the cervicodorsal vertebra on the body block. The boomerang-shaped furcula has an angle of approximately 80° and curves posteriorly in lateral view. No hypocleidium or posterior groove are evident, and the furcula lacks strong anteroposterior compression unlike most paravians besides *Velociraptor* (*Norell & Makovicky, 1999*:7). Remaining pectoral girdle elements include a portion of a strap-like scapula without a dorsal flange (preserved in part as natural mold on the body and skull blocks), and most of an enlarged coracoid, consistent with the pectoral girdle of other paravians. The coracoid possesses a posterior fossa and supracoracoid foramen but lacks signs of proximal pneumaticity.

The left forelimb is largely complete, missing the distal portion of the humeral shaft and the non-ungual phalanges of digits I and II; these elements are distributed across the skull, body, and hand blocks. The proximal half of the left humerus has been mechanically prepared to be fully free of matrix. The distal half of the right humerus is preserved on the body block, providing a nearly complete composite humerus. The ulna and radius are shorter than the humerus, and the resulting forearm is proportionately short (estimated forelimb to hindlimb ratio of 0.58, Table 1) as seen in *Mahakala*, *Mei*, *Tianyuraptor*, *Caihong*, *Zhenyuanlong*, *Austroraptor*, and *Halszkaraptor* (*Xu & Norell, 2004*: supp. table 4; *Novas et al., 2008*: table 1; *Zheng et al., 2009*: table 1; *Turner, Pol & Norell, 2011*: table 1; *Lu & Brusatte, 2015*: supp. info. 1; *Cau et al., 2017*: supp. table 1; *Hu et al., 2018*: supp. table 1), and was clearly incapable of flapping flight.

The humerus is significantly shorter than the femur (0.63) and slender. The deltopectoral crest is proximally restricted, roughly triangular, and projects closer to perpendicular to the head's long axis (Fig. 14E). *Hesperornithoides'* crest lacks the fenestra found in some microraptorians and the distinct lateral scar (Fig. 14F) seen in *Coelurus*, attributed to the m. pectoralis superficialis by *Carpenter et al. (2005b*:60, Fig. 3.8B*)*. The humeral head is anteriorly concave and proximally convex, and not separated from the bicipital crest by a capital groove. Well projected medially but unprojected anteriorly, this crest is proximodistally short but has a straight inner edge. There is no trace of a pneumatricipital fossa or foramen in the proximal humerus. Distally the humerus is well expanded and exhibits an enlarged entepicondyle, over 15% of distal humeral width (Fig. 14D). Among maniraptorans, this is otherwise only seen in avialans such as *Zhongjianornis*, *Sapeornis*, *Jixiangornis*, *Confuciusornis*, and various ornithothoracines (*Chiappe et al., 1999*: Fig. 38; *Zhou & Zhang, 2003*: Fig. 7a).

Unlike any of the three well preserved Morrison basal coelurosaurs, the distal ulna is highly compressed dorsoventrally to be over twice as wide as tall. It is also dissimilar from *Coelurus* and *Tanycolagreus* in being straight in side view (*Carpenter, Miles & Cloward, 2005a*: Fig. 2.10B-C; *Carpenter et al., 2005b*: Fig. 3.8C-D). The distal end lacks

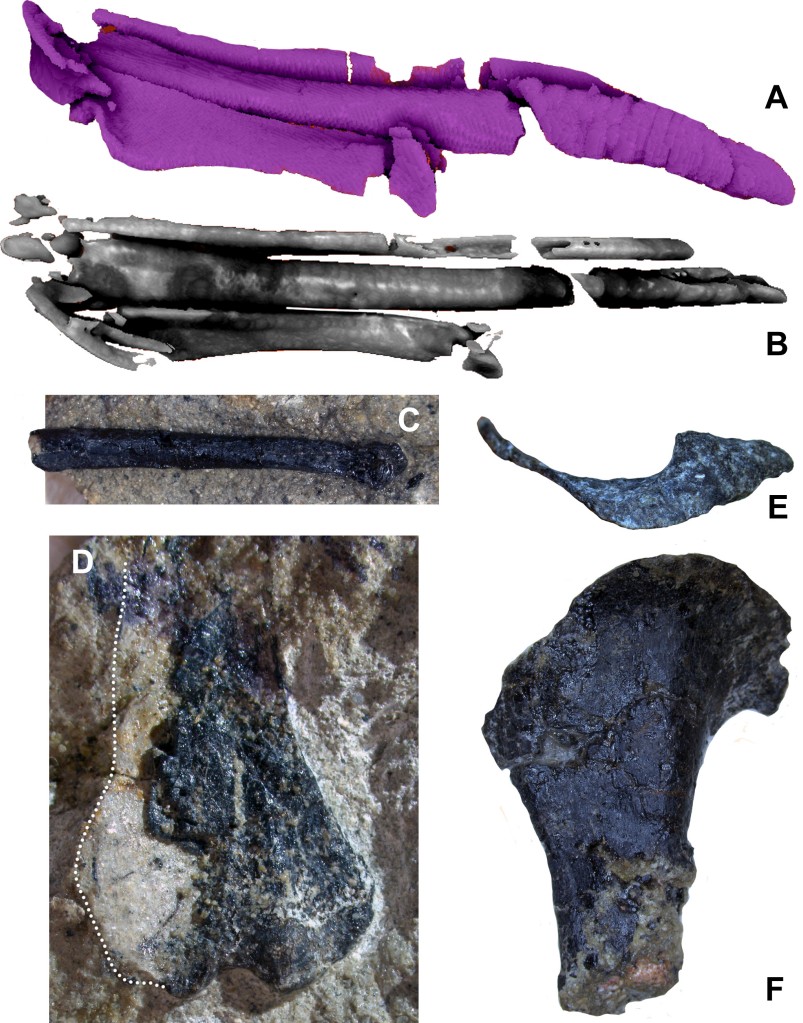

**Figure 14 Forelimb elements of WYDICE-DML-001.** Segmented left carpals and metacarpals in oblique (A) and extensor (B) views. (C) Distal portion of right MC III in lateral view. (D) Distal end of right humerus in anterior view. Proximal end of left humerus in proximal (E) and lateral views (F).

significant proximoventral development of the articular surface, is roughly straight in dorsal perspective and has no well-defined radial sulcus. The radius itself is over 70% of ulnar width at midshaft, exhibits no obvious groove or scaring posterodorsally on the shaft but does possess a distodorsal flange typical of pennaraptorans.

The semilunate carpal is preserved next to, but slightly displaced from the metacarpal packet on the skull block. It is not well exposed on the surface of the block, but CT scans reveal a well-developed semilunate morphology with a transverse trochlear groove (Fig. 14A). This is unlike the unfused distal carpal I of the much larger *Coelurus* specimen YPM 2010 which is more oblong than semilunate (*Carpenter et al., 2005b*: Fig. 3.9A). *Tanycolagreus* fuses the distal carpals but the resulting structure is very flat instead of semilunate (*Carpenter, Miles & Cloward, 2005a*: Fig. 2.11E-F; note the radiale is mistakenly identified as the semilunate). The size of the semilunate shows it covered most or all of

the proximal ends of metacarpals I and II. A well-developed mediodorsal process was present for articulation with the first metacarpal.

Metacarpal (MC) I is complete, while MC II–III are both missing the distal-most articular condyles; the left MCs can be seen on the right side of the skull block. An isolated distal MC III is also preserved from the other manus. Preserved phalanges are located on the hand block, with the exception of an isolated manual ungual. MC I is the shortest and most robust metacarpal (Fig. 14B), featuring an extensor flange as in paravians (*Gishlick, 2002*) but unlike *Ornitholestes* or *Tanycolagreus*. Its distal end is deeply ginglymoid, with the lateral condyle extending further ventrally. The metacarpals become progressively less robust laterally and there is no bowing of MC III. MC III is more robust than in *Tanycolagreus*, however, where it is much narrower than half the width of MC II (*Carpenter, Miles & Cloward, 2005a*: Fig. 2.12B). MC II exhibits a dorsal scar for the m. extensor carpi ulnaris longus equivalent to the intermetacarpal process in some Aves, which *Gishlick (2002)* recovered as exclusive to Pennaraptora.

The articulated digit III is exposed on the hand block in medial view, along with phalanx II-1 and manual ungual II in lateral view. Phalanx III-3 is longer than the expected combined lengths of III-1 and III-2. Manual unguals are large, raptorial, and trenchant. They have well-developed, proximally placed flexor tubercles and lack a proximodorsal lip (Fig. 15). An isolated ungual reported as an enlarged pedal ungual II by *Hartman, Lovelace & Wahl (2005)* is reinterpreted as manual ungual I, as the dorsal margin arches significantly above the articular facet when the latter is held vertically, and the large flexor tubercle extends significantly beyond the palmar side of the articular facet (cf. *Senter, 2007b*). Manual ungual III is subequal in size to ungual II (Figs. 15B and 15C) unlike *Tanycolagreus* (*Carpenter, Miles & Cloward, 2005a*: Fig. 2.12A) and most paravians except *Daliansaurus*, troodontid IGM 100/44 and *Mahakala* (*Barsbold, Osmolska & Kurzanov, 1987*: plate 50 Fig. 2-4; *Turner, Pol & Norell, 2011*: Fig. 29; *Shen et al., 2017b*: Table 1).

## Pelvic girdle and hind limb

An isolated block contains much of the ilial postacetabular process, partly as an impression that can nonetheless be reconstructed precisely via CT scans. Unlike the condition in *Ornitholestes* (*Carpenter et al., 2005b*: fig. 3.10A; note the ilium is photographed at a slight ventral angle and that the postacetabular process is blunt if viewed perpendicular to the blade—M. Mortimer, 2009, personal observation, AMNH 619), the postacetabular process is distally pointed in lateral view, has only a shallow brevis fossa and possesses a laterally projecting ventral lobe like some basal dromaeosaurids and troodontids. *Hesperornithoides* possesses a concavity along the dorsal edge of its postacetabular process. Originally considered an unenlagiine synapomorphy (*Makovicky, Apesteguía & Agnolín, 2005*:S15), the condition has proven to be widespread among theropods.

Hind limb elements are all preserved on the body block. The hind limbs are elongate relative to individual vertebrae, as in most coelurosaurs including *Archaeopteryx* and *Jeholornis* (*Dececchi & Larsson, 2013*). The left femur has been entirely prepared off of the block prior to reattachment, and is missing only the head and proximal-most portion; enough of the shaft is preserved to establish the absence of a fourth trochanter as in

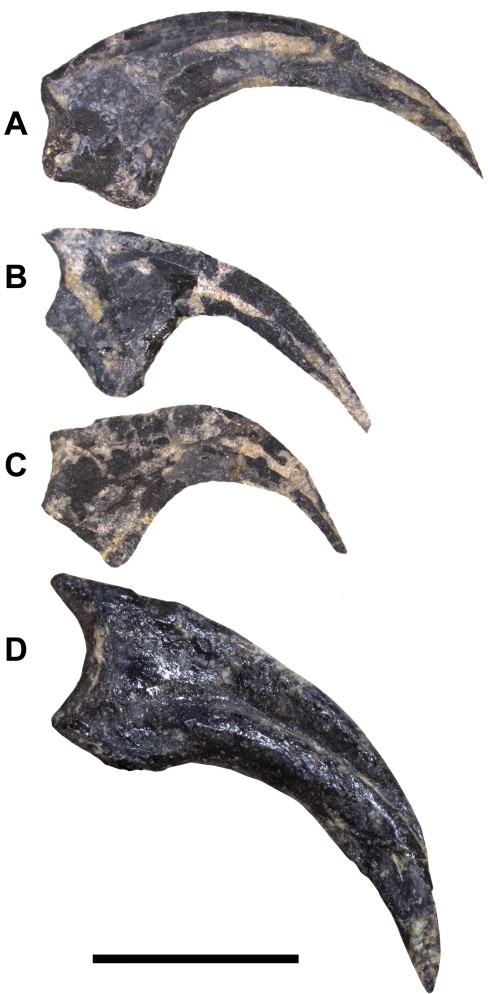

**Figure 15 Unguals of WYDICE-DML-001.** (A) Left manus ungual I in medial view. (B) Left manus ungual II in medial view. (C) Left manus ungual III in medial view. (D) Pedal ungual II (mirrored for ease of comparison) with trenchant sickle morphology. Scale bar = one cm.

the vast majority of pennaraptorans but unlike *Coelurus* and *Tanycolagreus* (*Carpenter et al., 2005b*:66; *Carpenter, Miles & Cloward, 2005a*: Fig. 2.14B). Notable among previously described Morrison paravian postcrania, proximal femur BYU 2023 (*Jensen & Padian, 1989*) only overlaps with WYDICE-DML-001 at midshaft where it also lacks a fourth trochanter. Our analysis recovered BYU 2023 as a deinonychosaur that could belong to a troodontid or dromaeosaurid *Hesperornithoides* without an increase in tree length (see Positions of maniraptoromorphs pruned a posteriori in the Supplementary Information), but further comparison is limited. Distally, our specimen lacks a significant extensor groove or medial epicondyle, and has a lateral condyle which is not projected distally and is separated from both the medial condyle and ectocondylar tuber.

The left tibia and fibula are exposed almost completely, with the anterior portion of the shafts buried in the the body block. Proximally the tibia is longer than wide, with a deep incisura tibialis and anteriorly projected cnemial crest that diverges from the shaft at a

high angle. No medial cnemial crest is developed. *Hesperornithoides* is similar to *Sinovenator*, the *Almas* holotype and *Achillobator* among paravians in having a lateral tibial condyle that extends anteriorly to overlap the incisura tibialis (*Perle, Norell & Clark, 1999*: plate XII Fig. 12C; *Xu et al., 2002*: Fig. 1h; *Pei et al., 2017b*: Fig. 9). This is unlike the condition in *Tanycolagreus* (*Carpenter, Miles & Cloward, 2005a*: Fig. 2.14K). The proximally placed fibular crest is separated from the proximal condyles and has a longitudinal groove on its posterior edge. Distally, the lateral malleolus is covered by the proximal tarsals but the medial malleolus is distally exposed.

The fibula extends the full length of the tibia but is extremely reduced in diameter. Most of the shaft is less than one millimeter at maximum thickness. Broken portions reveal the fibula lacks trabecular bone, being hollow and exceptionally thin walled. *Hesperornithoides* also differs from *Tanycolagreus* in that its fibula is subequal in transverse width anteriorly and posteriorly, whereas the latter genus exhibits a posterior tapering (*Carpenter, Miles & Cloward, 2005a*: Fig. 2.14K).

The astragalus and calcaneum are tightly appressed to the distal ends of the tibia and fibula but are not co-ossified to them or each other. The ascending process of the astragalus is elongate unlike *Coelurus* (*Carpenter et al., 2005b*: Fig. 3.12A), though narrow unlike most paravians except for *Anchiornis* and *Scansoriopteryx* (*Czerkas & Yuan, 2002*: Fig. 20; *Hu et al., 2009*: Fig. S4d). It is separated from the astragalar body by a transverse groove and contacts the fibula at its lateral edge. Similar to avialans and most troodontids (*Xu, 2002*: Fig. 47D; *Ji et al., 2011*: Fig. 3F; *Zanno et al., 2011*: Fig. 9D; *Xu et al., 2012*: Fig. 1K; *Brusatte et al., 2013*: 67) but in contrast to *Coelurus* and *Tanycolagreus* (*Carpenter et al., 2005b*: Fig. 3.12E; *Carpenter, Miles & Cloward, 2005a*: Fig. 2.14L), the astragalocalcanear anterior intercondylar groove is deep (over 20% of tarsal depth).

The left metatarsus is preserved in articulation with the zeugopod and is almost completely exposed on the body block. Distal tarsals are not fused to the metatarsals (MT), which are themselves also unfused. Although the left metatarsus is incomplete distally, it clearly displays a sub-arctometatarsalian condition (Figs. 16A, 16B and 16D) unlike *Tanycolagreus* and *Ornitholestes* (*Carpenter, Miles & Cloward, 2005a*: Fig. 2.15B; *Carpenter et al., 2005b*: Fig. 3.13A), with MT III being constricted along the shaft but not excluded in anterior view. The metatarsus is quite different from *Ornitholestes*, being slender and closely appressed with transversely compressed metatarsals (Fig. 16D; compare to Fig. 13C in *Holtz, 1994*). MT III further differs from all three well preserved Morrison basal coelurosaurs in being straight in proximal view instead of L-shaped (*Carpenter, Miles & Cloward, 2005a*: Fig. 2.15C; *Carpenter et al., 2005b*: Fig. 3.13A, D-based on medial margin of proximal metatarsal IV for *Coelurus*). Posteriorly, MT III is exposed as a narrow sliver along its proximal length. MT II is not slender as in derived troodontids, and it and MT III lack tubercles for the m. cranialis tibialis. Only a posterior portion of the distal metatarsus is mechanically exposed, and it is not well resolved in the CT data. MT I is straight with no torsion, and has a slightly constricted neck just before the incompletely preserved distal condyles (Fig. 16C). A slender MT V is preserved in articulation at the posterolateral edge of MT IV (Figs. 16A and 16D). Due to distal breakage it is uncertain if the element is elongated as in dromaeosaurids.

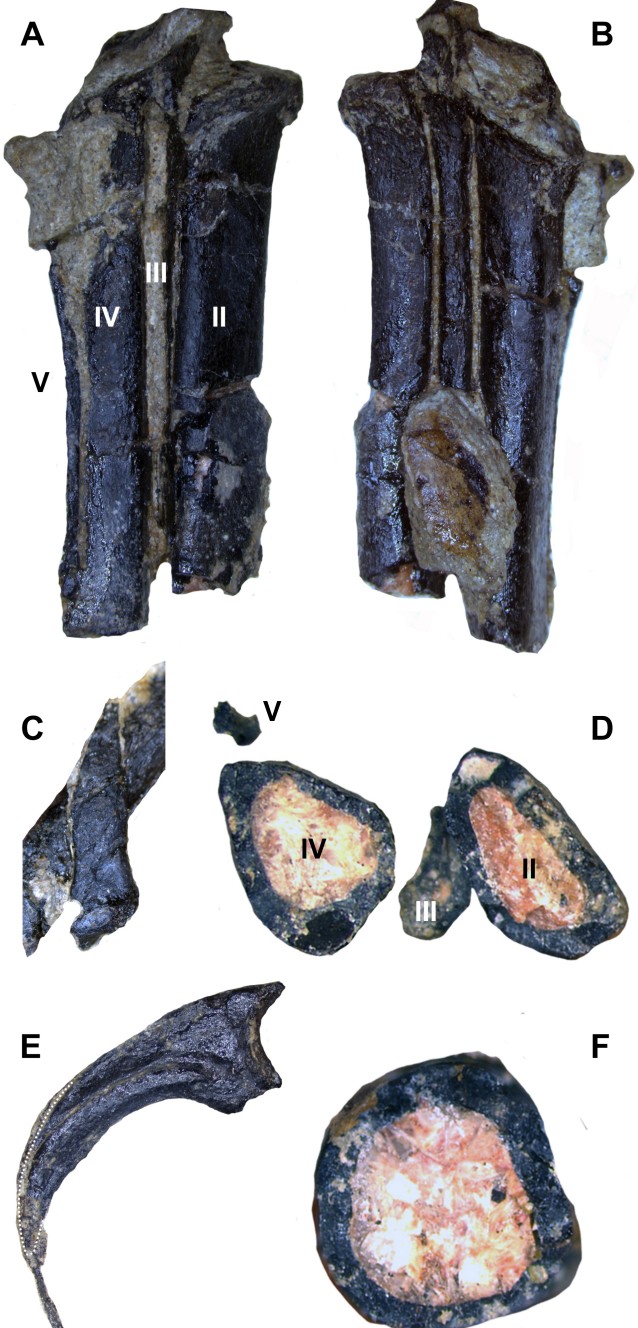

**Figure 16 Select hindlimb elements of WYDICE-DML-001.** Proximal end of left metatarsal packet in posterior (A) and anterior (B) views. (C) Right metatarsal I in medial view. (D) Cross-section through left metatarsals from distal perspective, anterior is to the bottom. (E) Enlarged pedal ungual II: material outside the dotted line is inferred to be preserved sheath material. (F) Cross-section through the left tibia at mid-shaft.                             

Left pedal digits III and IV are preserved in articulation with the plantar and lateral surfaces exposed at the base of the body block. As in *Archaeopteryx*, toes III and IV are subequal in length, proportions that appear less specialized for functional didactyly, with

digit IV being substantially shorter and less robust than digit III (*Mayr et al., 2007*: Fig. 13a). Pedal phalanx IV-4 is shorter than IV-3, associated with cursoriality more than arboreality or grasping. Proximal portions of pedal unguals III and IV are preserved; they are compressed in section, do not appear to be strongly curved and lack enlarged flexor tubercles. An isolated ungual was found that is strongly recurved and trenchant, exhibiting an intermediately expanded flexor tubercle (Fig. 15D). It differs from manual unguals in having a dorsal margin that does not arch above the articular facet when the facet is vertical. Given the differences from both the manual unguals and the articulated III and IV pedal unguals, we interpret this as an enlarged semi-raptorial pedal ungual II. It also preserves a filagree of organic material that is consistent with previously published remnants of keratin sheaths (Fig. 14E; *Schweitzer, 2011*).

# PHYLOGENETIC ANALYSIS

## Phylogenetic methods

Almost every large-scale coelurosaur analysis from the past two decades is ultimately a derivative of *Norell, Clark & Makovicky (2001)* and have come to be known as Theropod Working Group (TWiG) analyses. While TWiG can refer to the core group of AMNH affiliated researchers, we use it more inclusively to encompass the myriad analyses and derivatives based on their dataset (Fig. S1). This widespread adoption has made the TWiG dataset arguably the most successful lineage of dinosaur phylogenetic analyses to date. Most subsequent iterations add one to several taxa and a few to several characters to a preexisting version of the matrix. While this practice has resulted in character and taxon list stability between coelurosaur analyses, it has also led to endemic issues in the compilation of data matrices. *Jenner (2004)* identified similar concerns for metazoan cladistics, finding that taxon selection and sampling, uncritical copying of scores, and poorly formed character states have been overlooked and can compromise the validity of topologies recovered by phylogenetics programs that use these as their primary data. In light of this, rather than adding *Hesperornithoides* to an existing TWiG matrix we have instead begun an overhaul of the TWiG dataset to comply with modern ideals of data matrix construction.

Subsequent TWiG analyses and derivatives have not added characters and taxa in a single "lineage" of analyses, but instead have formed "clades" of analyses that increasingly diverge in content (see Fig. S1). So, for example, modern analyses derived from *Senter (2007a)* such as *Senter et al. (2012)*, and those derived from *Turner (2008)* such as *Brusatte et al. (2014)* have over a hundred characters not found in the other. A similar pattern occurs with taxonomic sampling; most new paravians are added individually or in small groups to a version of the TWiG matrix but these additions are frequently not propagated to subsequent analyses. To date no analysis has added all, or even most, newly discovered taxa to a single matrix. We follow the advice of *Jenner (2004)* that authors should attempt to include all previously proposed characters and terminal taxa, while explicitly justifying omissions. To this end we have attempted to include every character from all TWiG papers published through 2012, with the goal to continually add characters

from other analyses in future iterations. Each excluded character from these studies is justified under Excluded Characters in the Supplementary Information.

We have also scored almost every named Mesozoic maniraptoromorph known from more than single elements or teeth (the seven exceptions are noted under Excluded Taxa in the Supplementary Information), as well as 28 unnamed specimens. Five recent examples of Aves were included, the palaeognath *Struthio* and the neognaths *Chauna*, *Anas*, *Meleagris*, and *Columba*. The Tertiary *Lithornis* and *Qinornis* were also included as both have been suggested to be outside Aves by some authors, as were *Palaeotis*, *Anatalavis*, *Presbyornis*, *Sylviornis*, *Gallinuloides*, *Paraortygoides*, and *Foro* as basal representatives of modern clades. Historically, TWiG analyses have focused on coelurosaurs while using *Sinraptor dongi* and *Allosaurus* as outgroups. As some taxa have been alternatively recovered as ceratosaurs or coelurosaurs (e.g., *Deltadromeus*, *Afromimus*), we tested the current character list against an exhaustive sample of Mesozoic averostrans with *Dilophosaurus* as the outgroup. This enabled us to test the content of Maniraptoromorpha and provided us with a more representative outgroup sample than merely *Sinraptor* and *Allosaurus*. However, while the coelurosaur characters utilized do recover many traditional clades outside Maniraptoromorpha, the topology of this section should not be viewed as well supported since characters specific to ceratosaurs, megalosauroids, and carnosaurs are lacking.

One issue is that many TWiG analyses (a notable exception being Senter (2007a)) reuse scorings from prior analyses they were derived from, even when additional data has been published. This can perpetuate errant scores and fails to account for more recent discoveries, publications, and interpretations. As an example, *Harpymimus* has been scored identically for only three cranial characters in every TWiG analysis not derived from Senter (2007a) since Norell, Clark & Makovicky's (2001) original up through Turner, Makovicky & Norell (2012). Even analyses which focused on basal ornithomimosaurs (e.g. Makovicky et al., 2010) sometimes failed to utilize Kobayashi & Barsbold's (2005) redescription of *Harpymimus* for updated scorings. We address this by reexamining and rescoring each character for every taxon based on direct personal examination, high resolution photographs, and/or literature published through 2018.

Perhaps more importantly, we have set out to improve character selection and construction. It is often unappreciated that accurate character scores are only useful if the characters and states themselves are objective, independent and formed so that phylogenetics programs will correctly interpret proposed homology. Existing TWiG state definitions are often unquantified, which hinders future authors from scoring taxa consistently or objectively determining accuracy of published scores. Composite characters, which score multiple variables in a single character, are also common. Correlated characters have increasingly become an issue in analysis lineages as authors add data from other analyses that are not independent of characters already in their matrix. Other issues we addressed include eliminating the use of "absent" as a state in a transformational character (Sereno, 2007:582–584), character traits constructed with discontinuous quantified states (so that certain ranges of values aren't covered by any state) and those that include a state merely scoring for any condition except those specified

by the other states (*Jenner, 2004*:301–302). We have begun the process of resolving these issues by quantifying 163 characters, isolating 240 composite states into single variables (often using the other variables to form new characters), and excluding 36 correlated characters (see Excluded Characters in the Supplementary Information). Our character list includes details of how each character has been changed from previously published versions. When possible, newly quantified character states have been formulated to best match the taxon distribution for each originally subjective character. All characters have been rewritten in the logical structure advocated by *Sereno (2007)* to reduce ambiguity and variability between analyses.

The resulting matrix includes 700 characters and 501 OTUs. A total of 10 characters are parsimony-uninformative among our 389 maniraptoromorphs (excluding the possibly tyrannosauroid megaraptorans, coelurids, and proceratosaurids in this and the following taxon totals, to ensure similar content). These are retained pending future expansions of the analysis, leaving 690 parsimony-informative characters among our taxon sample of maniraptoromorphs. This makes it the second largest character sample and the largest taxonomic sample in a TWiG analysis of maniraptoromorphs to date, compared to other recent iterations of each TWiG lineage- *Gianechini et al. (2018)* (700 parsimony-informative characters for their 135 maniraptoromorph OTUs), *Foth & Rauhut (2017)* (534 such characters and 120 such OTUs), *Brusatte et al. (2014)* (666 such characters and 127 such OTUs), *Agnolin & Novas (2013)* (405 such characters and 80 such OTUs), and *Senter et al. (2012)* (367 such characters and 98 such OTUs).

The data matrix was analyzed with TNT 1.5 (*Goloboff & Catalano, 2016*). After increasing the "Max. trees" under "Memory" in "Settings" to 99999, an initial "New Technology search" using "Sect. Search," "Ratchet," "Drift" and "Tree fusing" was run, as a "Driven search" to "Find min. length" 15 times. The level was checked every three hits, and Sect. Search settings were changed to "For selections of size . . ." " . . . above" 45 and RSS settings to "Factor for number of selections" 84 and "Max. sector size" 267. This search was stopped after 100 hours at a length of 12,132 steps. A series of "New Technology search"es using "Sect. Search" (with "CSS" unchecked), "Ratchet," "Drift" and "Tree fusing" were run from "RAM" which located trees 12,128 steps in length. Constraint analyses were run using a file containing one shortest tree, unchecking "Settings" "Lock trees," checking "Trees" "View" and moving OTUs (left click, right click) to fulfill the constraint. An initial "Traditional search" of "trees from RAM" with 100 "Max. trees" and "enforce constraints" was then run, followed by a "New Technology search" with 10,000 "Max trees" using "Sect. Search" (with "CSS" unchecked), "Ratchet," "Drift" and "Tree fusing" from "RAM" using "Enforce constraints." These traditional and new technology searches were alternated until the length was unchanged for three searches. Constraint analysis results which were shorter than 12,128 steps were combined as the only trees in a new file which was itself analyzed as a "New Technology search" with 99999 "Max trees" using "Sect. Search" (with "CSS" unchecked), "Ratchet," "Drift" and "Tree fusing" from "RAM" until no shorter trees were recovered. A "Traditional search" of "trees from RAM" with default parameters was then performed to more fully explore the final treespace.

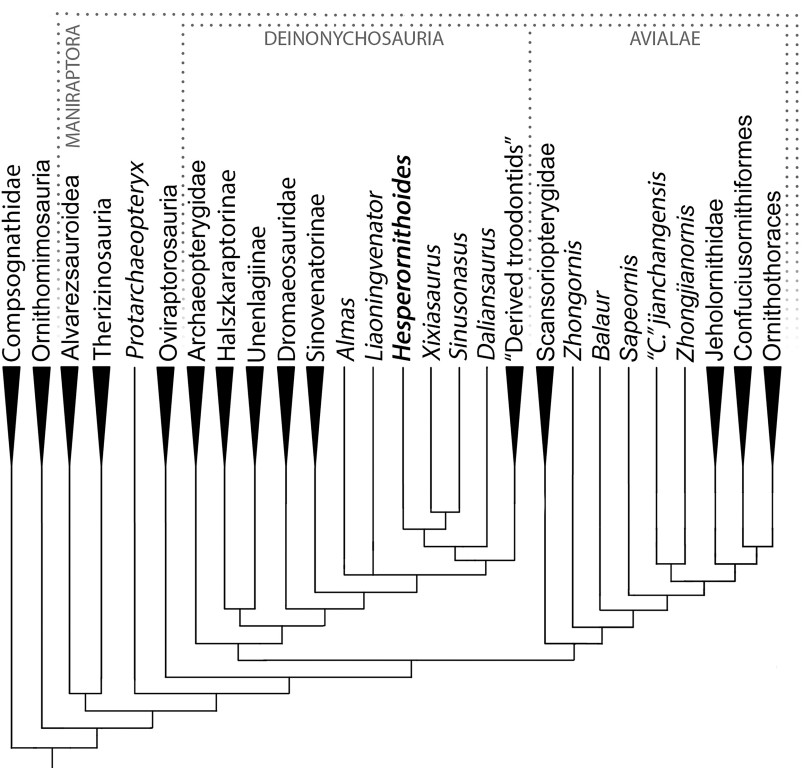

**Figure 17 Summary diagram of the findings from this phylogenetic analysis.** Strict consensus tree of maniraptoromorphs after a posteriori pruning with higher level taxa condensed (length = 12,123). The uncondensed tree and positions of pruned taxa can be seen in the Supplemental Data.

## Phylogenetic results

The analysis resulted in >99999 most parsimonious trees with a length of 12,123 steps. The recovered trees had a consistency index of 0.073, and a retention index of 0.589. Figure 17 presents a summary of the strict consensus tree after the a posteriori pruning of several taxa with multiple possible positions (see Supplementary Information for complete results). Megaraptorans and a clade of proceratosaurids and coelurids branch first after tyrannosauroids, yet both groups are often recovered as members of the latter clade in analyses sampling more characters relevant rootward of Maniraptoromorpha. Indeed, megaraptorans or the coelurid-proceratosaurid group can be constrained to Tyrannosauroidea in only four steps, while it takes 10 steps to move the next closest taxon to birds, *Ornitholestes*. As *Ornitholestes* has never been recovered as a tyrannosauroid it is considered the most basal well supported member of Maniraptoromorpha here.

Compsognathids emerge closer to maniraptoriforms than *Ornitholestes*, and several taxa usually considered members of that family (e.g., *Huaxiagnathus*, *Juravenator*, *Mirischia*, *Sinocalliopteryx*) branch off more basally. As all previously suggested characters (*Peyer, 2006*:880; *Brusatte, 2013*:559) connecting these taxa to *Compsognathus* were utilized, our increased data sampling supports the more reduced Compsognathidae recovered here. This version of the family includes several controversial taxa. As described

in the Supplementary Information, *Sciurumimus'* placement away from *Compsognathus* in the analyses used by *Rauhut et al. (2012)* was due almost entirely to misscorings in the original versions of those analyses and only one of its characters unexpected in a basal maniraptoromorph is not used in our analysis. As it requires seven steps to move to Megalosauroidea, the compsognathid identification is better supported. *Aorun* was recently recovered as an alvarezsauroid by *Xu et al. (2018)* based on four characters, one of which we include (our 29) and another (proximodistal oblique ridge on tibia bracing astragalar ascending process) seemingly present in *Compsognathus* as well (*Peyer, 2006*: Fig. 10; scored unknown by Xu et al.). Constraining it to be an alvarezsauroid in our matrix adds 17 steps, so seems to be highly unlikely. Similarly, *Haplocheirus* falls out in Compsognathidae instead of in its traditional position as a basal alvarezsauroid. Eight characters used by *Choiniere et al. (2010)* to place it in the latter clade were not included, but it also requires nine steps to constrain there in our analysis even with the inclusion of the supposedly intermediate *Xiyunykus* and *Bannykus*. This suggests neither a compsognathid nor an alvarezsauroid identification is well supported and more study is needed.

Another area with less support than suggested by consensus is the base of Maniraptoriformes, where we recover alvarezsauroids and therizinosaurs as the first branching maniraptorans as in most recent studies. However, only four steps are required to get a result similar to *Sereno's (1999)* where alvarezsauroids are sister to ornithomimosaurs and therizinosaurs sister to that pair. Similarly, while we recover a pairing of alvarezsauroids and therizinosaurs to the exclusion of pennaraptorans, placing therizinosaurs closer to the latter clade merely needs three additional steps. Positioning alvarezsauroids sister to Pennaraptora or putting therizinosaurs just outside Maniraptoriformes are slightly less parsimonious at six steps each, but the once popular topology of a therizinosaur-oviraptorosaur clade is much less likely at 13 steps longer.

Among ornithomimosaurs, *Deinocheirus* and the odd *Hexing* form the first branching clade unlike *Lee et al. (2014a)* where the former is well nested sister to *Garudimimus*. As we use all valid characters from that analysis placing *Deinocheirus* close to *Garudimimus* and *Beishanlong*, and it takes 14 steps to constrain that result, it is here rejected. However, the Mongolian giant can be placed within the toothless ornithomimosaur clade using merely four additional steps, so its basal position may change as the referred material is more fully described. Early Cretaceous American *Arkansaurus* and *Nedcolbertia* are both resolved as ornithomimosaurs for the first time. Few characters have been proposed to organize taxa within Ornithomimosauria and many taxa lack detailed descriptions or justified alpha taxonomy. Thus, our topology should be regarded as tentative (e.g., the placement of *Harpymimus* within toothed ornithomimosaurs can be changed with only two steps), but the consensus pairing of *Anserimimus* with *Gallimimus* and *Struthiomimus* with *Dromiceiomimus* requires eight more steps, so deserves additional scrutiny.

Therizinosauria retains standard relationships among basal taxa, with *Falcarius*, *Jianchangosaurus*, *Beipiaosaurus*, and *Alxasaurus* successively closer to therizinosaurids. *Martharaptor* has several equally parsimonious positions as a therizinosaur less closely related to *Therizinosaurus* than *Alxasaurus*, matching the position recovered by

*Senter et al. (2012)*. The topology is similar to the latest major study, *Zanno's (2010)* TWiG analysis whose data was fully utilized, in that *Suzhousaurus*, *Erliansaurus*, and *Neimongosaurus* are outside a clade including *Nanshiungosaurus*, *Nothronychus* spp., *Erlikosaurus*, and *Segnosaurus*.

*Fukuivenator* has a poorly specified position at the base of Maniraptora, emerging as the first branching alvarezsauroid, but moving to a basal therizinosauroid position with only two steps. A more stemward position seems more likely than a relationship with dromaeosaurids as suggested in its original description or *Cau (2018)*, as it can be a coelurid with only four more steps, but takes seven steps to be sister to Pennaraptora and 11 steps to be paravian. The next branching alvarezsauroid is *Nqwebasaurus*, as also recovered by *Dal Sasso & Maganuco (2011)*, which requires six steps to be an ornithomimosaur or seven steps to be closer to Pennaraptora. All but two characters recovered by *Choiniere, Forster & De Klerk (2012)* as supporting an ornithomimosaurian placement were included, so the alvarezsauroid alternative is stronger. A compsognathid position as in *Novas et al. (2012)* is 10 steps longer so even less likely. Rounding out the controversial basal ornithomimosaurs is *Pelecanimimus,* recovered as the first branching alvarezsauroid based on characters such as an elongate anterior maxillary ramus (311:1), posterior tympanic recess in the otic recess (27:1), lateral teeth set in grooves (302:1), over 30 dentary teeth (90:3), and a proximally expanded metacarpal II (370:0). Constraining it as an ornithomimosaur only requires two additional steps, however, where it emerges just above *Shenzhousaurus* as in *Macdonald & Currie (2018)*. As only two of their characters supporting an ornithomimosaurian identification were not used by us, and only one from *Brusatte et al. (2014)*, its true position is unclear pending a detailed osteology such as *Perez-Moreno's (2004)* unreleased description.

As in *Xu et al.'s (2018)* new analysis, *Patagonykus* and *Bonapartenykus* are outside Alvarezsauridae, but unlike that study *Xiyunykus* and *Bannykus* join them as patagonykines. Alvarezsaurid *Patagonykus* requires four steps, and Xu et al.'s new genera follow, while placing the new genera more stemward than *Patagonykus* only takes two steps. Thus, neither of these arrangements should be viewed as heavily favored until Xu et al.'s taxa are described in detail. Within Parvicursorinae, not all characters from *Longrich & Currie's (2009)* alvarezsaurid analysis were included, but we recovered a clade including only *Mononykus*, *Shuvuuia*, *Parvicursor* and the Tugrik taxon as in derivatives of that study. *Heptasteornis* is recovered as an alvarezsaurid, supporting *Naish & Dyke (2004)*. Note while the controversial *Kinnareemimus* fell out as a tyrannosauroid in the shortest trees, only three steps move it to Ornithomimosauria and four to Alvarezsauroidea. Similarly, while *Kol* resolves as a relative of *Avimimus* as suggested by *Agnolin et al. (2012)*, a single step places it in Alvarezsauridae.

*Protarchaeopteryx* emerges as the sister group of Pennaraptora, but changes to the first branching oviraptorosaur with merely two steps and the first branching paravian with three steps, suggesting no strong signal for this poorly preserved specimen. Placing it in Archaeopterygidae as originally proposed and supported by *Paul (2002)* requires 11 additional steps, however, which strongly outnumber the few valid published characters for such an arrangement. Oviraptorosaurs include *Similicaudipteryx* as their first

branching member and unusually places *Incisivosaurus* closer to caenagnathoids than *Caudipteryx* and within Oviraptoridae itself. It only requires a single step to make *Incisivosaurus* the sister taxon of Caenagnathoidea, and three steps to make the first branching oviraptorosaur, so any of these positions are plausible. Forcing *Incisivosaurus* and *Protarchaeopteryx* to be sister taxa as in *Senter et al. (2004)* requires six steps, and the duo resolves as the first branching oviraptorosaur clade. Within Caenagnathoidea, our results should be considered incomplete pending incorporation of characters from *Maryanska, Osmolska & Wolsan (2002)* and its derivatives. Notable similarities to (*Lee et al., 2019*) include *Nomingia* as an oviraptorid and a clade of *Nemegtomaia* and *Heyuannia*. Major differences include non-caenagnathoid *Ganzhousaurus* (one step needed to make oviraptorid; emerges sister to *Heyuannia*), caenagnathid *Avimimus* (three steps to place outside Caenagnathoidea) and *Machairasaurus* (two steps needed to make oviraptorid; emerges basalmost), and oviraptorid *Microvenator* (one step needed to make caenagnathid, three steps to move it outside Caenagnathoidea), *Gigantoraptor* (four steps needed to make caenagnathid; emerges basalmost), and *Beibeilong* (two steps needed to make caenagnathid; second branching after *Microvenator*). Maryanska et al.'s heterodox hypothesis of avialan oviraptorosaurs requires 12 additional steps despite the inclusion of proposed intermediates such as *Epidexipteryx* and *Sapeornis*. As we use a far greater maniraptoromorph taxon and character sample (*Maryanska, Osmolska & Wolsan (2002)* includes 16 such taxa and 162 parsimony-informative characters), and only lack two characters they use to support avialan oviraptorosaurs, the traditional content for Paraves is significantly more parsimonious.

A Deinonychosauria including troodontids and dromaeosaurids was recovered as in many recent analyses. Positioning troodontids closer to Aves than dromaeosaurids only requires a single additional step, but non-eumaniraptoran troodontids are less parsimonious at six more steps. Scansoriopterygids form the first branch of Avialae, matching their stratigraphic placement, and constraining them as basal paravians instead is only one step longer. Their other suggested position as oviraptorosaurs requires 12 more steps though, so is unlikely. While *Pedopenna* emerges as a scansoriopterygid in the MPTs, one step moves the fragmentary specimen to Archaeopterygidae instead. The juvenile *Zhongornis* branches next, with alternative positions in Scansoriopterygidae or Confuciusornithiformes being four and five steps longer respectively. *Balaur* follows and only moves to Dromaeosauridae with eight additional steps, supporting its placement in Avialae by *Cau, Brougham & Naish (2015)*. The branching order of Jehol non-ornithothoracine birds has been contentious, with our matrix supporting *Sapeornis* branching first, followed by jeholornithids then confuciusornithiforms. Jeholornithids branching first is only three steps longer, but *Sapeornis* branching last as in some recent analyses requires 12 more steps. Note *Changchengornis* moves one node to Confuciusornithiformes in merely two steps and *Jinguofortis* joins *Chongmingia* in only three steps. Our analysis supports the latter's position close to Ornithothoraces as in p2 of *Wang et al.'s (2016)* figure 7, whereas moving it to their p1 more stemward of *Jeholornis* and *Sapeornis* requires 11 more steps.

Characters supporting enantiornithine monophyly and phylogeny are not strongly sampled, making this portion of the tree provisional. Despite this, several proposed clades

were recovered including Pengornithidae, *Liaoningornis* plus *Eoalulavis*, *Sinornis* plus *Cathayornis*, and Longipterygidae with *Longipteryx* outside a clade containing *Longirostravis*, *Shanweiniao*, and *Rapaxavis*. One proposed clade which is strongly rejected is Bohaiornithidae, requiring 16 additional steps to make monophyletic using *Wang et al.'s (2014)* taxonomic content. Among controversial taxa, *Evgenavis* and *Qiliania* can move to Confuciusornithiformes using only three steps and one step respectively, *Liaoningornis* and *Hollanda* can move to Ornithuromorpha in four steps each, and *Vorona* requires six steps to move to that clade. The proposed pairings of *Aberratiodontus* and *Yanornis* (*Zhou, Clarke & Zhang, 2008*) and *Ambiortus* and *Otogornis* (*Kurochkin, 1999*) are unparsimonious at eight and 17 additional steps. Among taxa closer to crown Aves, the grade of taxa stemward of *Bellulornis* are usually placed in enantiornithines and can move there in one to five steps depending on the OTU. Thus, their position may be revised given the inclusion of more enantiornithine characters, whereas, for example, *Bellulornis* and *Archaeorhynchus* would take 14 and 17 steps to move to Enantiornithes respectively so are solidly closer to Aves.

Our analysis includes all of Clarke's lineage of ornithuromorph characters (originating from *Clarke, 2002*) as incorporated into *Turner, Makovicky & Norell's (2012)* TWiG analysis, so should be a good test for this portion of the tree. *Bellulornis* is the sister taxon to Ornithuromorpha, followed by a pairing of *Archaeorhynchus* and *Patagopteryx* as in *Zheng et al. (2018)*. Placing the latter genus closer to carinates than *Gansus* as in some recent analyses requires nine more steps. Songlingornithidae is recovered, but Hongshanornithidae forms a grade and requires 10 more steps to constrain with *Wang et al.'s (2015)* taxonomic content. With *Field et al.'s (2018)* new *Ichthyornis* data included, it falls out sister to hesperornithines, but the alternatives with either toothed taxon being closer to Aves only need four additional steps each. *Eogranivora* and the poorly described *Xinghaiornis* form the first branching carinates, followed by *Iaceornis* and *Apsaravis*. *Lithornis* is just outside Aves, but can be a palaeognath in five additional steps. While the character list was not designed to resolve Aves, consensus clades are largely recovered, including the recently recognized Vegaviidae and an anseriform *Teviornis*.

Incorporating the new data from *Rauhut, Foth & Tischlinger (2018)* on *Archaeopteryx* and *Pei et al. (2017a)* on *Anchiornis* nests the former genus in the clade usually called Anchiornithinae, making this entire group Archaeopterygidae. As in *Xu et al. (2011)* we recover archaeopterygids as deinonychosaurs, but both the traditional *Archaeopteryx* position closer to Aves and the common *Anchiornis* position sister to troodontids require a single additional step each. Even making archaeopterygids sister to dromaeosaurids requires merely four more steps, but placing them on the paravian stem as *Lefèvre et al. (2017)* recovered for anchiornithines is 15 steps longer. As only two of their characters supporting this stemward placement were unused by us, that position is rejected here. Among complete archaeopterygids *Caihong* is notably labile and can be a dromaeosaurid with only two more steps, given its mesially serrated teeth and unreduced distal caudal prezygapophyses.

As in *Senter et al. (2012)* and *Cau (2018)* our trees pair unenlagiines and halszkaraptorines, but uniquely places this Unenlagiidae sister to the dromaeosaurid plus troodontid clade.
Unenlagiids can take their traditional position sister to other dromaeosaurids in trees one step longer where archaeopterygids pair with troodontids, or can be placed closer to Aves as in *Agnolin & Novas (2013)* in trees one step longer where archaeopterygids are in this position as well. *Mahakala* is only weakly connected to *Halszkaraptor*, becoming a basal paravian in one step, but eight new characters proposed as halszkaraptorine synapomorphies by *Cau et al. (2017)* were not used so this may be an artifact. *Hulsanpes* emerges as a dromaeosaurine in the shortest trees, but only takes two steps to move to Halszkaraptorinae. The briefly described taxon *Ningyuansaurus* resolves as the sister taxon to *Mahakala* but becomes a basal paravian in one more step and an oviraptorosaur as originally proposed in four steps. It should be reexamined, but characters such as the low iliofemoral ratio (261:1), low ischiopubic ratio (187:3) and enlarged pedal ungual II (224:1) are more like paravians than oviraptorosaurs. For the first time the European *Pyroraptor* and *Ornithodesmus* are recovered as unenlagiines, which would match biostratigraphically with the presence of traditionally Gondwanan clades such as spinosaurids, carcharodontosaurids and abelisaurs in the Cretaceous of Europe. Giant *Dakotaraptor* also falls out in this group instead of sister to *Dromaeosaurus* as in *DePalma et al. (2015)*, but the latter study did not provide their scorings for the genus although we used all their characters. Moving it to Dromaeosauridae only takes three steps, but it resolves as a basal taxon instead of a eudromaeosaur. The unenlagiine *Rahonavis* had previously been recovered by *Agnolin & Novas (2013)* and trees based on Cau's analysis as closer to Aves in a similar position to *Jeholornis*. We recover it nested in Unenlagiinae, and it takes 10 additional steps to move closer to Aves than archaeopterygids and other unenlagiines. We analyzed every character supporting this in Agnolin and Novas' matrix, so the unenlagiine consensus seems strong, especially as four of their characters connecting *Rahonavis* with more derived birds are correlated with having long wings.

*Tianyuraptor* is recovered as the most basal dromaeosaurid but can be placed in Microraptoria with one step and closer to eudromaeosaurs in two steps. Similarly, *Zhenyuanlong* is a microraptorian in the shortest trees, but can be nearer to eudromaeosaurs in two steps as well. Constraining the two to be sister taxa to simulate the synonymy suggested by *Makovicky, Gorscak & Zhou (2018)* is only four steps longer, and the pair emerge on the eudromaeosaur branch. We recover *Bambiraptor* as an early branching microraptorian as in *Senter et al. (2004)*, with *Variraptor* as its sister taxon. However, *Bambiraptor* moves closer to eudromaeosaurs as in *Senter et al. (2012)* and *Cau et al. (2017)* with the addition of a single step, and *Variraptor* can join Unenlagiinae with the other European taxa in only three steps. Fragmentary *Yurgovuchia* is sister to Eudromaeosauria and requires four steps to place close to *Utahraptor* and *Achillobator* as in *Senter et al. (2012)* despite using all of their characters supporting this. *Deinonychus* joins *Utahraptor* and *Achillobator* to form a large dromaeosaurid clade not previously hypothesized, which is most parsimoniously outside Dromaeosaurinae plus Velociraptorinae, but can be moved to either subfamily in two steps. This group also includes the controversial *Yixianosaurus*, which takes seven steps to move to a more stemward position in Maniraptora as in *Dececchi, Larsson & Hone (2012)*, 10 steps to be a basal paravian as in

*Foth, Tischlinger & Rauhut (2014)*, and 13 steps to place by scansoriopterygids as originally proposed by its describers. Another new coalition is a Dromaeosaurinae including *Saurornitholestes, Atrociraptor, Tsaagan, Linheraptor*, and *Itemirus*, though the first two were recovered as close relatives by *Longrich & Currie (2009)* and *Tsaagan* and *Linheraptor* have been proposed to be synonymous by several authors and were resolved as sister taxa by *Cau et al. (2017)*. Our results agree with most recent studies in placing *Adasaurus* in Velociraptorinae, along with the newly analyzed *Luanchuanraptor* and unnamed Djadochta specimen IGM 100/980. Although *Acheroraptor* and *Velociraptor? osmolskae* resolve as microraptorians, a single step nests the former in Eudromaeosauria, and two steps joins the latter to *Velociraptor mongoliensis*, so these jaw-based taxa are not strong evidence of Late Cretaceous microraptorians. Conversely, the Campanian *Hesperonychus* holotype emerges as an avialan at least as close to Aves as *Balaur* despite all of its microraptorian-like characters being used. Three steps are needed to constrain it to Microraptoria. *Agnolin & Novas (2013)* uniquely proposed microraptorians to be in Avialae (under their junior synonym Averaptora), but as we included all their characters supporting this arrangement and still find it takes 14 additional steps to constrain, we strongly reject the hypothesis.

The recently named Sinovenatorinae are the first branching troodontids, including not only *Mei* and the eponymous *Sinovenator*, but also *Xiaotingia, Jianianhualong*, and unnamed IGM 100/140 described by *Tsuihiji et al. (2015)*. Troodontid characters present in *Xiaotingia* but not anchiornithines include distally positioned obturator process (183:2), and characters shared with sinovenatorines include large posterior surangular foramen (80:2), capital groove in humerus (458:1), metacarpal III extending distally past metacarpal II (640:1), laterally ridged ischium (182:2), and enlarged pedal ungual II (224:1). Forcing *Xiaotingia* into Archaeopterygidae requires nine more steps, which strongly suggests it is not a member considering we included all of the TWiG data originally used to place it there. Alternative placements as a non-anchiornithine avialan (*Lee et al., 2014b*), a dromaeosaurid (*Senter et al., 2012*), and a scansoriopterygid relative (*Lefèvre et al., 2017*) are eight, five, and 26 more steps, respectively. *Jianianhualong* and IGM 100/140 were originally recovered as closer to troodontines by their describers, and both can move there with a single step. One node closer to troodontines are *Almas*, possibly referrable perinates IGM 100/972 and 100/974, and a clade of *Liaoningvenator* and unnamed Ukhaa Tolgod specimen IGM 100/1128. *Almas* and IGM 100/1128 were recovered as jinfengopterygines by *Turner, Makovicky & Norell (2012)* but the eponymous *Jinfengopteryx* has a highly unstable position in our analysis, equally capable of joining with these taxa or falling out in Sinovenatorinae. Even a position in Archaeopterygidae as in *Ji et al. (2005)* is only three steps longer, and a basal paravian position as in *Foth, Tischlinger & Rauhut (2014)* is just four steps longer. In comparison, moving *Liaoningvenator* to group with anchiornithines as in *Shen et al. (2017b)* requires 10 steps. *Zanabazar, Linhevenator, Talos*, and *Troodon* sensu lato form a derived group of troodontines, with *Saurornithoides, Urbacodon, Gobivenator*, and *Byronosaurus* successively more distant sister taxa. The classic Early Cretaceous specimen IGM 100/44 branches first in the final troodontid clade, with *Daliansaurus* and a pairing of *Xixiasaurus* and *Sinusonasus*

successively closer to our new taxon *Hesperornithoides*. *Daliansaurus* and *Sinusonasus* were recovered as sinovenatorines by *Shen et al. (2017a)* and can be constrained there in four and six steps, respectively.

Finally, constraint analyses were used to test alternative placements for *Hesperornithoides*. In order to quantify the likelihood of it being a juvenile *Ornitholestes*, *Coelurus*, or *Tanycolagreus*, we constrained trees pairing *Hesperornithoides* with each Morrison OTU. These were 11, 15, and 16 steps longer, respectively, than the most parsimonious trees, corroborating the abundant character evidence described above that *Hesperornithoides* is not referable to a Morrison non-maniraptoriform. While not unique, troodontid synapomorphies such as the pneumatic quadrate (60:1), anterior cervical centra which extend posterior to their neural arches (104:1), and the deep tibiotarsal intercondylar groove (206:1) place *Hesperornithoides* within that clade. Characters like the small dental serrations (92:1), elongate but not hypertrophied distal caudal prezygapophyses (127:1), straight ulna (367:0), dorsally projected curve on manual ungual I (378:1), and enlarged manual ungual III (391:1) are homoplasic but combine to position the new taxon with *Daliansaurus*, *Xixiasaurus*, and *Sinusonasus*. Yet like its contemporary *Archaeopteryx*, *Hesperornithoides* can easily move to different positions in the paravian tree. A placement as the first branching dromaeosaurid is just two steps longer, supported by the dorsally placed maxillary fenestra (321:1), mesial dental serrations (89:0), and large lateral teeth (91:0). This may be more compatible stratigraphically, but moving *Hesperornithoides'* clade to a more stemward position in Troodontidae outside Sinovenatorinae, the *Liaoningvenator*-like taxa and derived troodontids is also only two steps longer. Similarly, in trees two steps longer than the MPTs where troodontids are avialans, *Hesperornithoides* can be the first branching taxon closer to Aves than troodontids based on homoplasic characters such as the short posterodorsal lacrimal process (44:0/1). Two additional steps also place the taxon in contemporaneous Archaeopterygidae, sister to *Caihong* which shares character states 89:0, 91:0, and 127:1. Despite the uncertainty of its position within Paraves, however, *Hesperornithoides* is strongly supported as a member of the Deinonychosauria plus Avialae clade, as even constraining it to the paravian stem requires 15 additional steps.

## DISCUSSION

*Hesperornithoides miessleri* adds another small-bodied theropod to the list of dinosaur taxa from the well-studied Morrison Formation (*Foster, 2003*), reinforcing the importance of continued exploration and excavation of well-sampled formations. Regardless of its position within Paraves, *Hesperornithoides* is significant given the previous lack of Jurassic troodontids or dromaeosaurids known from non-dental remains, or of Jurassic avialans and/or archaeopterygids from the Americas. If it is a troodontid as the most parsimonious trees suggest, it would establish the presence of multiple species in the Jurassic of North America in conjunction with *K. douglassi* (though see *Holtz, Brinkman & Chandler, 1998*).

Our phylogenetic analysis and constraint tests suggest the apparent in TwiG derivatives may often be one of multiple equally plausible alternatives, from the topology of early

maniraptoriform clades to the structure of Paraves. Indeed, a single step separates several different paravian phylogenies including such heterodox concepts as an archaeopterygid-troodontid sister group and a pairing of troodontids and dromaeosaurids exclusive of unenlagiids. Yet this does not mean anything goes, as multiple proposed topologies were rejected by our data. These include alvarezsauroid *Aorun*, paravian *Fukuivenator*, deinocheirid *Garudimimus*, the therizinosaur-oviraptorosaur clade, avialan oviraptorosaurs, oviraptorosaurian scansoriopterygids, *Yixianosaurus* or *Xiaotingia* sister to scansoriopterygids, basal paravian archaeopterygids, avialan microraptorians, *Rahonavis* closer to Pygostylia than archaeopterygids or unenlagiines, pygostylian *Sapeornis*, and Bohaiornithidae with its original content. This realm of plausibility has not always been made obvious in prior TWiG analyses, as few explicitly test alternative topologies. When alternatives are tested, the likelihood of their reality may be understated such as when *Turner, Makovicky & Norell (2012)* reported as we do that only one step is necessary to recover troodontids in Avialae instead of Deinonychosauria. Yet they still stated "Deinonychosaurian monophyly is well supported in the present cladistic analysis and has been consistently recovered in all TWiG analyses after the original *Norell, Clark & Makovicky (2001)* analysis." This illustrates the importance of viewing cladograms as a network of more or less likely relationships instead of a new "correct" topology. Given the propensity of authors to reuse previous scorings and character constructions, repeated phylogenetic results may result in a false impression of confidence not justified by constraint analyses. We urge authors going forward to be vigilant in checking old character scorings, to formulate uncorrelated and quantifiable new characters scoring for single variables when expanding past analyses, and to check alternative topologies' strength.

Even considering the range of parsimonious maniraptoran topologies, our phylogenetic results provide important observations on the origin of avian flight. Basal maniraptoran clades such as alvarezsauroids and therizinosaurs are unambiguously non-volant. Short-armed *Protarchaeopteryx* lies near the divergence of oviraptorosaurs and paravians, while basal oviraptorosaurs exhibit a grade of short-armed basal taxa including *Similicaudipteryx*, *Caudipteryx*, and *Avimimus* (Table 3). Within Paraves we find unambiguously non-flying taxa at the base of all clades regardless of topology. *Archaeopteryx* (humerofemoral ratio 112–124%) is nested within shorter-armed Tiaojishan taxa (78–104%) that lack feathers adapted for advanced aerodynamic locomotion (*Saitta, Gelernter & Vinther, 2018*; *Pan et al., 2019*) whether *Caihong* is an archaeopterygid or a dromaeosaurid. Halszkaraptorines are short-armed, and if *Mahakala* and/or *Ningyuansaurus* are basal paravians instead these support the hypothesis even further. Long-armed *Rahonavis* is deeply nested in Unenlagiinae, as is *Microraptor* within Dromaeosauridae even if *Tianyuraptor*, *Zhenyuanlong*, and *Bambiraptor* are allowed several steps to be closer to eudromaeosaurs. All troodontids have humeri 70% or less of femoral length with the exception of *Xiaotingia* (Table 3), including the early *Hesperornithoides* if it is a member. Moving the latter to Dromaeosauridae or Avialae would only cement the pattern further.

This pattern of short-armed basal members of paravian outgroups and subclades (Fig. 18) is important for understanding the timing of avian flight acquisition, as

**Table 3  Humeral/femoral ratios of paravian theropods.**

| Taxa | H/F ratio | Taxa | H/F ratio |
|---|---|---|---|
| *Ningyuansaurus* | 0.56 | *Deinonychus* | 0.76–0.80 |
| *Protarchaeopteryx* | 0.70 | *Austroraptor* | 0.47 |
| *Similicaudipteryx* | 0.59 | *Buitreraptor* | 0.91 |
| *Caudipteryx* | 0.47 | *Unenlagia* | 0.72 |
| *Mahakala* | 0.50 | *Dakotaraptor* | 0.57 |
| *Mei* | 0.52–0.55 | *Archaeopteryx* | 1.12–1.24 |
| *Sinovenator* | 0.68 | *Serikornis* | 0.90 |
| *Xiaotingia* | 0.85 | *Anchiornis* | 0.96–1.04 |
| *Jianianhualong* | 0.70 | *Eosinopteryx* | 0.78 |
| *Jinfengopteryx* | 0.70 | *Aurornis* | 0.88 |
| *Liaoningvenator* | 0.59 | *Halszkaraptor* | 0.60 |
| *Sinornithoides* | 0.59 | *Scansoriopteryx* | 1.06–1.12 |
| *Hesperornithoides* | 0.56 | *Yandangornis* | 0.75 |
| *Tianyuraptor* | 0.65 | *Epidexipteryx* | 0.98 |
| *Caihong* | 0.59 | *Zhongornis* | 1.04 |
| *Zhenyuanlong* | 0.63 | *Zhongjianornis* | 1.48 |
| NGMC 91 | 0.92 | *Sapeornis* | 1.57–1.88 |
| *Changyuraptor* | 0.97 | *Jeholornis* | 1.40–1.47 |
| *Sinornithosaurus* | 0.92 | *Jixiangornis* | 1.34–1.56 |
| *Microraptor* | 0.79–0.92 | *Eoconfuciusornis* | 1.11–1.30 |
| *Zhongjianosaurus* | 0.73 | *Chongmingia* | 1.18 |
| *Bambiraptor* | 0.85 | *Confuciusornis* | 1.14–1.27 |

**Note:**
 Data drawn from personal measurements and literature values.

individuals with humerofemoral ratios of 70% or less lack the wing-loading to generate significant horizontal or vertical thrust (*Dececchi, Larsson & Habib, 2016*). This contradicts hypotheses of neoflightlessness in oviraptorosaurs (*Maryanska, Osmolska & Wolsan, 2002*; *Feduccia & Czerkas, 2015*) or paravians in general (*Paul, 2002*), and also contradicts plesiomorphically volant dromaeosaurids (*Xu et al., 2003*). The recently discovered *Yi* suggests that at least some scansoriopterygids developed a divergent, parallel form of aerial locomotion (*Xu et al., 2015*), but given the significant epidermal and morphological differences between *Yi*'s forelimb anatomy and avian wings it seems most likely to have occurred independently of avian flight. This holds true whether scansoriopterygids are early branching avialans or basal paravians. The most parsimonious interpretation of our results is a series of parallel appearances of non-avian aerodynamic locomotion within microraptorians, unenlagiids, archaeopterygids, and scansoriopterygids.

Traditional attempts to understand the origin of avian flight have centered on the use of well-known, supposedly intermediary taxa such as *Archaeopteryx* and *Microraptor* to serve as key evolutionary stages (*Ostrom, 1979*; *Xu et al., 2003*). Our results (Fig. 18) suggest that whatever aerial locomotion these taxa may have engaged in, they did not give rise to avialan flight. Models of avian flight origins based on these taxa may be

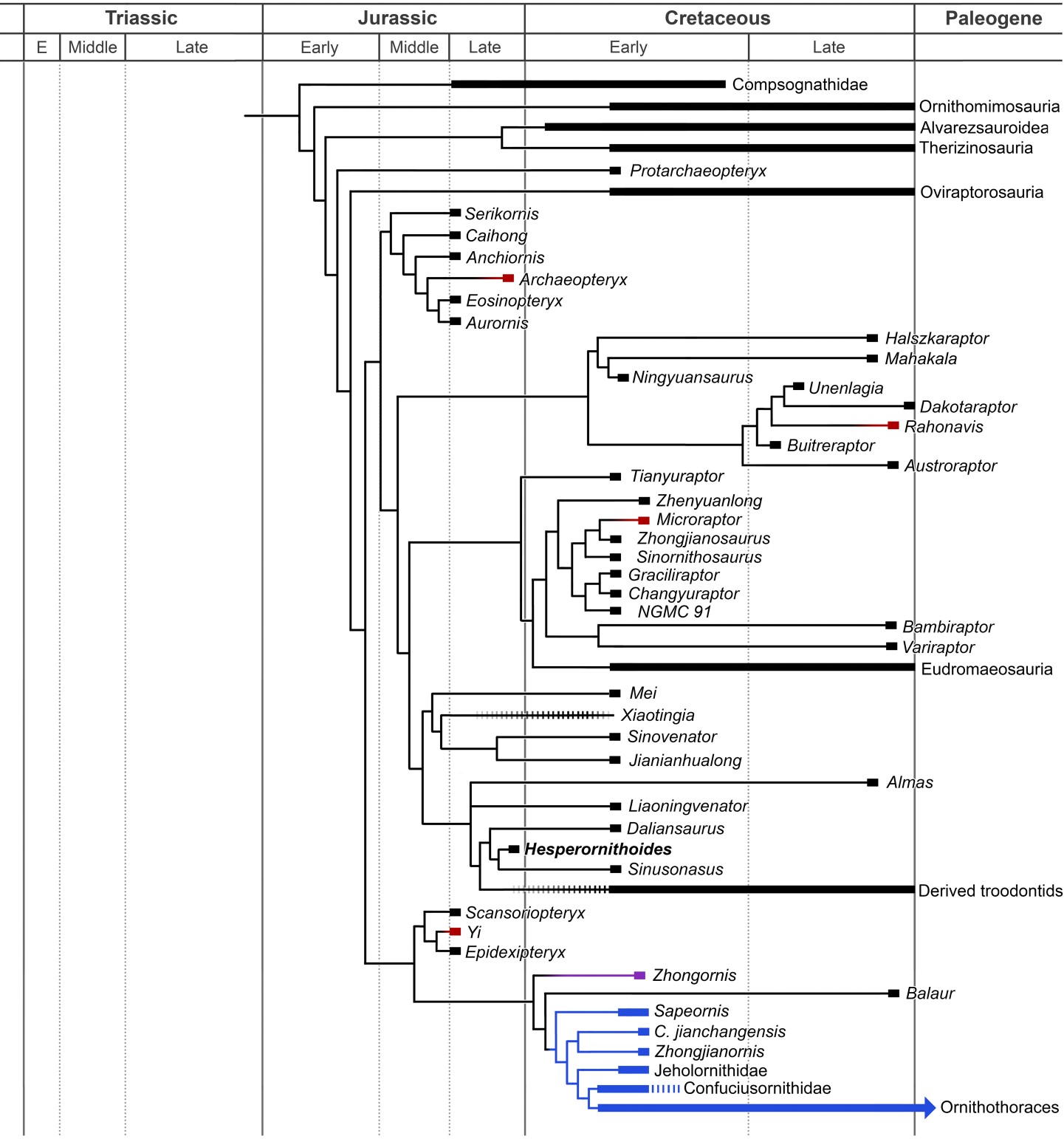

**Figure 18 Partially expanded, time calibrated phylogenetic results.** Clades containing potentially volant taxa (red) are expanded to show their position nested within flightless taxa (black). Taxa exhibiting aerial locomotion directly connected to crown clade Aves are colored blue. *Zhongornis* is colored purple to reflect the uncertainty revolving around this juvenile specimen. Barred lineages indicate uncertainty in age (*Xiaotingia*) or referred taxa (*Koparion* to troodontids and non-Jehol taxa to Confuciusornithiformes).

misinterpreting the sequence of character acquisition that resulted in crown avian flight. The embedding of putatively flighted *Rahonavis, Archaeopteryx* (though see *Agnolin et al., 2019*), and *Microraptor* within clades that lack evidence of aerial locomotion is consistent with prior studies that found the morphology of most non-avialan paravians as functionally more similar to terrestrial birds and mammals than arboreal ones (*Dececchi & Larsson, 2011*; *Agnolin et al., 2019*). This supports a non-volant terrestrial ecomorph as the basal condition for the major paravian clades, supporting numerous previous studies demonstrating that key flight preadaptations up to and including vaned feathers and well-developed wings evolved in terrestrial contexts millions of years prior to the origin of crown avian flight (*Makovicky & Zanno, 2011*; *Brusatte et al., 2014*; *Dececchi, Larsson & Habib, 2016*; *Cau, 2018*).

With the morphologically divergent and potentially volant *Yi* (*Xu et al., 2015*) nested within a non-flying clade of basal avialans, one counter-intuitive result is that even avialans may have been plesiomorphically flightless. Though not a novel hypothesis (*Ostrom, 1979*; *Speakman & Thomson, 1994*; *Dececchi & Larsson, 2011*), it suggests the possibility for a surprisingly late acquisition of avian flight. Though new Jurassic fossils have the potential to push the origin of avian flight deeper in time, at the moment our first branching preserved examples are the Early Cretaceous *Zhongjianornis, Sapeornis*, and possibly *Zhongornis* (Fig. 18). Investigating the differences in flight capabilities and mode of life between them, confuciusornithids and more crownward avialans may be the most fruitful line of inquiry for understanding the transition to true avian flight.

The pattern of character acquisition, adult body reduction size (*Benson et al., 2014*), and parallel emergences of aerodynamic locomotion within Paraves suggests one possible solution to the traditional dichotomy of arboreal vs. terrestrial habitats in the origin of avian flight. While short-armed, non-arboreal alvarezsauroids, oviraptorosaurs, troodontids, and dromaeosaurs demonstrate that wings and other key characters associated with avian flight evolved in a terrestrial context, it is notable that clearly volant avialans like *Zhongjianornis*, and clades with putatively aerial behavior such as microraptorians, some archaeopterygids, and scansoriopterygids exhibit the strongest evidence among paravians for arboreal or semi-arboreal behavior. This suggests a model wherein small size and increasing approximations of the flight stroke allowed some clades of terrestrial paravians to utilize wing assisted incline running to access trees or other subvertical substrates previously not accessible (*Tobalske & Dial, 2007*; *Dececchi, Larsson & Habib, 2016*). From there the utility of gliding or flap-descent (*Norberg, 1985*; *Rayner, 1988*) provides a logical selective pressure that could generate several parallel experiments with aerial behavior, only one of which led directly to avian flight.

## CONCLUSIONS

We have described *Hesperornithoides miessleri*, a new paravian theropod from the Late Jurassic of North America. We ran a phylogenetic analysis based on previous TWiG datasets with expanded taxonomic sampling and recovered it as a troodontid, the oldest diagnostic specimen from North America known from more than teeth. *Hesperornithoides* was clearly a non-volant, terrestrial theropod that spent at least a portion of its life in a

marginal lacustrine or wetland environment. The terrestrial and flightless lifestyle is consistent with the base of Paraves, and with the base of paravian subclades, suggesting that avian flight evolved within Avialae, most likely in the Late Jurassic or Early Cretaceous.

## ACKNOWLEDGEMENTS

We would like to thank Howard and Helen Miessler for their support and generosity. We also thank volunteers from both the Tate Geological Museum and the Wyoming Dinosaur Center who aided the excavation. Thanks are due to Levi Shinkle for additional photography of WYDICE-DML-001 and to Dan Chure for supplying photographs of *Koparion*. High resolution CT scans were carried out at UT-Austin, with additional CT work provided by the UW-Madison WIMR and Hot Springs County Memorial Hospital. We would like to thank Alexander Averianov, Hebert Bruno Campos, Andrea Cau, Gareth Dyke, Federico Gianechini, Michael Habib, Jaime Headden, Rutger Jansma, Zhiheng Li, Heinrich Mallison, Phil Senter, Lindsay Zanno, and others who provided unpublished data on specimens, and the AMNH staff for allowing access to their collections. A final thanks is due to Oliver Rauhut and an anonymous reviewer, whose feedback greatly improved the manuscript.

### Funding

This work was supported by the Jurassic Foundation, the Western Interior Paleontological Society, and by donors through Experiment.com. The funders had no role in study design, data collection and analysis, decision to publish, or preparation of the manuscript.

### Grant Disclosures

The following grant information was disclosed by the authors:
Jurassic Foundation.
Western Interior Paleontological Society.
Donors through Experiment.com.

### Competing Interests

The authors declare that they have no competing interests.

### Author Contributions

- Scott Hartman conceived and designed the experiments, performed the experiments, analyzed the data, contributed reagents/materials/analysis tools, prepared figures and/or tables, authored or reviewed drafts of the paper, approved the final draft.
- Mickey Mortimer conceived and designed the experiments, performed the experiments, analyzed the data, contributed reagents/materials/analysis tools, prepared figures and/or tables, authored or reviewed drafts of the paper, approved the final draft.
- William R. Wahl conceived and designed the experiments, performed the experiments, contributed reagents/materials/analysis tools, authored or reviewed drafts of the paper, approved the final draft.

- Dean R. Lomax conceived and designed the experiments, performed the experiments, authored or reviewed drafts of the paper, approved the final draft.
- Jessica Lippincott contributed reagents/materials/analysis tools, authored or reviewed drafts of the paper, approved the final draft.
- David M. Lovelace conceived and designed the experiments, performed the experiments, analyzed the data, contributed reagents/materials/analysis tools, prepared figures and/or tables, authored or reviewed drafts of the paper, approved the final draft.

### New Species Registration

The following information was supplied regarding the registration of a newly described species:

Publication LSID: urn:lsid:zoobank.org:pub:6325E8D2-0AAF-4ECD-9DF2-87D73022DC93.

*Hesperornithoides* LSID: urn:lsid:zoobank.org:act:DA4A267F-28C1-481E-AC06-DD7E39D0036F.

*Hesperornithoides miessleri* LSID: urn:lsid:zoobank.org:act:583400B9-AD76-42B3-AD01-D596FEBD2C57.

### Data Availability

All 3D PDFs are available at FigShare:

Hartman, Scott (2019): Skull block.pdf. figshare. Figure. https://doi.org/10.6084/m9.figshare.7029284.v1.

Hartman, Scott (2019): Body Block.pdf. figshare. Journal contribution. https://doi.org/10.6084/m9.figshare.7029299.v1.

In addition to the FigShare data, we collected data from the following specimens. All specimens described or visited for phylogenetic scoring are accessioned in recognised museum collections: American Museum of Natural History (AMNH), Mongolian Institute of Geology (IGM), Geological Museum of China (NGMC), Wyoming Dinosaur Center (WYDICE).

AMNH 619, AMNH 3041, AMNH 6515, AMNH 6516, AMNH 6558, AMNH 6565-6570, AMNH 6576, AMNH 7517, AMNH 21786-21803, AMNH 21626-21627, AMNH 21884-21892, AMNH 30240A-G, AMNH 30556, IGM 100/99, IGM 100/975, IGM 100/986, IGM 100/1128, IGM 100/1276, IGM 100/1323, NGMC 2124, NGMC 2125, NGMC 97-4-A, NGMC 97-9-A, WYDICE-DML-001 (holotype of *Hesperornithoides miessleri*, gen. et sp. nov.).

### Supplemental Information

Supplemental information for this article can be found online at http://dx.doi.org/10.7717/peerj.7247#supplemental-information.

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
