# Peer review of "A new paravian dinosaur from the Late Jurassic of North America supports a late acquisition of avian flight"

_PeerJ, doi:10.7717/peerj.7247_

## Round 0.1 · original submission · Major Revisions

Neither of the reviewers, nor the handling editor, denies the tremendous importance of this fantastic new find. However, the manuscript in its current form does not do it justice. Both reviewers raise major issues concerning the phylogenetic analysis in addition to a host of smaller issues. Please consider and respond to the comments point-by-point.

Reviewer 1 ·

Basic reporting

summarized below

Experimental design

poor-summarized below

Validity of the findings

poor-summarized below

Additional comments

This is a frustrating paper. The dinosaur being described is clearly a new species, and looks to be quite important, as a Jurassic-aged paravian fairly close to the origin of birds. There are, however, several major issues with this paper.

First, why is the description so short? This specimen looks to be one of the best skeletons of a Jurassic paravian (at least from outside China), and has been CT scanned. There is so much anatomical information, and comparisons with other paravians, that could be presented. But the description reads like something that was prepared for Science or Nature and unchanged for this submission. This is PeerJ. There is basically unlimited space. The authors should describe this fossil so that nobody needs to ever describe it again.

Second, and most critical, the phylogenetic analysis is bizarre, and in my opinion unsuitable for publication. The authors in essence begin with an outdated version of published dataset, the Theropod Working Group dataset (TWiG is the proper acronym used by the workers in this group, not TWG as used by the authors here) and extensively modify it according to their own personal style, producing a highly unorthodox phylogenetic tree. The authors are also highly critical of the Theropod Working Group project—a 20+ year iterative study of coelurosaur phylogeny and evolution that has produced larger and more synoptic datasets over time—saying that including their new species in the latest TWiG dataset would ‘perpetuate problems’, and that the TWiG project does not ‘comply with modern ideals of data matrix construction.’ The main reason given is that the TWiG team has not ‘attempt(ed) to include all previously proposed characters and terminal taxa’ (i.e., has not included every taxon and every character published by other, non-TWiG studies). As a result, the TWiG studies are said to present ‘a false impression of confidence not earned by the data.’

These are bold statements. The authors seem to misunderstand the TWiG project. They say that TWiG analyses subsequent to the initial publication of Norell et al. 2001 ‘have not added characters and taxa in a single “lineage” of analyses, but instead have formed a branching tree of analyses that increasingly diverge in content’. In fact, the core TWiG team based at the American Museum of Natural History has produced a series of analyses that build on each other by adding characters and taxa over time, as is well documented in the literature (Norell et al. 2001 to Turner et al. 2007 to Turner et al. 2012 to Brusatte et al. 2014 to the more recent versions of Pei et al. 2017 and Gianechini et al. 2018). Other authors, like Senter, have used the TWiG dataset as a template for their own analyses, but they are not part of the TWiG project. If Senter’s characters, or others, have diverged from the TWiG core team’s dataset, that is not because of shoddy work or sloppy divergence, it is a sign of a healthy field that is analyzing, reanalyzing, and building upon a standard dataset.

But all this is beside the point. Let’s look at what the authors have actually done with their phylogenetic analysis. This, not their sanctimonious tone, is where the real concerns lie.

The authors take a six-year-old version of the TWiG matrix (Turner et al. 2012) and then extensively redefine characters and change scores, based mostly on photographs and published literature. They have apparently seen very few specimens first-hand, which is an issue because the TWiG dataset was constructed almost entirely on two decades, and thousands of hours, of first-hand specimen study by the American Museum team. The version of the TWiG dataset they use is only about half the size of the most recent, updated version (Brusatte et al. 2014 and the small additions of Pei et al. 2017 and Gianechini et al. 2018), and thus is missing a wealth of key characters and taxa. The 2014 version has been out for four years—why haven’t the authors used it?! That makes no sense to me. The authors also extensively rewrite character statements to follow Sereno’s (2007) guidelines, seemingly unaware that Brusatte et al. (2014) reframed many of the characters in the same way. They justify revamping the dataset by cherry picking a few characters that they find problematic, namely one concerning the manual unguals that they find to be insufficiently quantified, and another few characters they consider to be non-independent. Fair enough: in a dataset of 800+ characters, not all of them will be written perfectly clearly, and some character non-independence will probably sneak through. These are small things that need to be revised, well within the norm for large datasets of morphological characters that have to be chosen and defined by a human brain and not a genetic sequencer. This is not some fatal flaw of the TWiG dataset. The authors’ new characters can suffer from same of the same issues.

So let’s be clear what they authors are doing: they are taking an outdated version of a standard dataset that has been poured over by a huge team of experienced researchers over the past 20 years, nitpicking about some small issues of character definitions, and then completely blowing it up and reforming it based on their own liking, mostly based on photographs, having seen very few original specimens.

Simply, for me, this just isn’t going to cut it. I don’t have faith in this approach, and I don’t have faith in the phylogenetic results, which are *highly* unorthodox.

This is a brief sample of some of the very unusual results of this analysis, some of which break with years of convention—not just in the TWiG analyses, but from numerous other studies by independent research groups. Pelecanimimus as an alvarezsauroid. Troodontids outside Eumaniraptora. Therizinosauroids more basal than ornithomimosaurs. Xiaotingia as a sinovenatorine. Halszkaraptor as an avialan. A non-maniraptoriform Haplocheirus. The list could go on. Maybe some of these relationships are correct. Maybe the authors are visionaries and everybody else—from Gauthier through to the TWiG team and dozens of other international teams—has been using badly-constructed characters or too-incomplete datasets. But I doubt it. It’s more likely that the authors’ analysis is flawed. I can’t in good faith allow the authors to blow up theropod phylogeny with this analysis, particularly as they are using their results to support a fairly unorthodox interpretation of the origin of flight.

It’s interesting that the authors dismiss some of their more unusual results by suggesting that these relationships may change as new characters are incorporated. Characters relating to alvarezsaurids outlined by Choiniere et al. (2010) were not included, characters relating to oviraptorosaurs published over 15 years ago by Maryanska et al. (2002) were not included, and of course, the enormous chunk of new TWiG data added by Brusatte et al. (2014), Pei et al. (2017), and Gianechini et al. (2018) were not included, much of which refers to tyrannosauroids and basal coelurosaurs, which have clear implications for the relationships of some of the clades found in unusual places in the authors’ analysis (ornithomimosaurs, therizinosauroids, alvarezsauroids). I agree, I think the exclusion of some of these characters may explain some of the weird results. That’s fine, not every analysis can include every single character that’s ever published. But it’s awfully brazen of the authors to heavily criticize the TWiG dataset for not including ‘all previously proposed characters’, when they themselves have excluded a bucketload of characters, including ones that they themselves admit are probably central to testing some of the weird results they find. Pot kettle black, I must say.

So what should the authors do? In my view, they cannot proceed with their current, half-baked attempt to blow up the TWiG matrix based on partial data and little examination of specimens. Clearly their new specimen is important and needs to be described, so I think there are two options if the authors want to retain a phylogenetic analysis in their paper:

1) Simply include the new taxon in the most recent version of the TWiG analysis. I suspect the authors won’t want to do this, because they seem to feel that the TWiG analysis is so flawed that it is useless. But this paper is primarily describing a new taxon. The authors don’t have to revise all of coelurosaurian phylogeny with it. They can save that battle for another day.

2) Step up to the plate and practice what they preach. Instead of taking an outdated version of the TWiG dataset and massaging it to their liking, start fresh and actually examine the many alternative datasets that are out there (TWiG, Senter, Cau, Foth & Rauhut, Agnolina & Novas, etc.), pool ALL of the characters and taxa together in the way they demand of others, come up with a proper comprehensive de novo phylogenetic analysis, and then publish a thorough revision of coelurosaur phylogeny and systematics.

It’s really only one or the other here.

I know my review seems harsh. But I’m trying to be fair and totally honest. Like many of my colleagues, I’ve been spending years (decades now even) slowly, gradually, carefully improving our knowledge of coelurosaur phylogenetics by working with other scientists to build ever-more inclusive datasets that are carefully vetted (by each other) and are based on meticulous observation of specimens. We probably are wrong about some things. Goodness knows the TWiG dataset has been producing different results over time as the dataset has grown. I’m not averse to change. But I am averse to disrespectful, incomplete efforts like the current bizarre phylogenetic analysis.

This new specimen is amazing. It deserves to be published. If done right, I could even see a Nature or Science paper here. But I’m sorry to say the authors are hitching their horse (and what a horse it is—a prize thoroughbred!) to a rickety wagon with rotten wheels.

·

Basic reporting

The manuscript presents a new theropod specimen from the Morrison Formation that represents a small coelurosaur, which is described as a new genus and species of dromaeosaurid. This would represent the oldest record of dromaeosaurids currently known. In general small theropods from the Morrison Formation are still very poorly known, but potentially have great importance for our understanding of early coelurosaur evolution. Thus, any new information on these animals is greatly appreciated.
The manuscript is generally well written and follows the usual scientific structure used in our field. The pertinent literature is included and has been used extensively.

Experimental design

As outlined above, the manuscript fits the aims and scope of PeerJ. The phylogenetic analysis includes a wealth of information and seems to be generally expertly carried out.

Validity of the findings

Although any new information on Morrison theropods is welcome, as stated above, the manuscript in its current form has some serious shortcomings that need to be addressed by the authors before it can be further considered for publication.
The first main problem concerns the description of the new taxon, which is so brief and poorly illustrated that it is almost impossible to evaluate the claims of the authors. The description basically consists of comments on a few selected characters and comparative statements, but it is not even clearly stated how much and which elements of the skeleton are preserved in the new specimen.
With the current description, I am not convinced that the specimen represents a paravian theropod, nor even that it necessarily is a new taxon. Please note that I am not saying that it is not, but the information provided is insufficient to verify these claims (none of the elements described or figured show clear maniraptoran synapomorphies, and the proximal part of the humerus actually seems to lack a previously proposed maniraptoran synapomorphy, a proximodistally elongate, rectangular internal tuberosity). A more thorough description of the actual morphology of the preserved elements (and not only some selected characters of a few elements) is necessary. Furthermore, I was surprised that basically no comparisons with other known small-bodied Morrison theropods, including Ornitholestes, Coelurus and Tanycoelagreus, are presented; as these would be the most likely candidates for a taxonomic identification if the specimen does not represent a new taxon, comparative statements of how this specimen differs from these taxa are absolutely necessary. This is especially the case for Coelurus, as the comparable elements (based on the description and few anatomical illustrations presented) actually show quite a few marked similarities. Thus, for example, the dentary referred to Coelurus is very similar to the dentary of the new specimen, as are the presacral vertebrae, and the humerus of Coelurus has a similarily extensive entepicondyle. Likewise, the skull shows similarities with that of Ornitholestes in having a pronounced lateral ridge at the lower boundary of the antorbital fossa and an anteroposteriorle elongate, kidney-shaped maxillary fenestra in the maxilla and a small pneumatic recess in the jugal. However, none of these similarities are commented on, nor are any characters mentioned that might help to differentiate the new specimen from these taxa.
Furthermore, in some instances, the description and the illustrations do not seem to match. Thus, in the description of the maxilla, the authors state that the maxillary fenestra is displaced dorsally if compared to other theropods. However, according to the figure, the fenestra is basically in line with the ventral margin of the antorbital fenestra, which is the same condition as in most other theropods. Another example is that the text states that a labiolingual constriction is present in the dentary teeth, referring to figure 6G. However, this figure shows a CT rendering of a tooth in lingual view, in which a labiolingual constriction would not be visible anyway. In their comparisons for this character, the authors furthermore refer to a figure that shows a tooth of Microraptor that clearly has a mesiodistal constriction, so I guess that they might refer to such a feature. However, a marked mesiodistal constriction between crown and root is clearly absent in the tooth figured in fig. 6G. Likewise, the description states that the presacral vertebrae lack external pleurocoels, but both the figure of the cervical vertebra in lateral view (fig. 8B) as well as the cervicodrosal vertebra (fig. 8F) clearly show large, matrix-filled depressions exactly in the position where I would expect an external pleurocoel.
Furthermore, in the absence of histological studies, I think the evidence that this specimen represents an adult is rather poor. The authors cite fused neural arches and partially fused cervical ribs as evidence for the adult status of the specimen. However, in small theropods, unfused, but largely closed neurocentral sutures can be difficult to identifiy (e.g. none of the known specimens of Archaeopteryx shows clearly identifiable neurocentral sutures, although basically all of these specimens are considered to be juveniles), and the two photos of presacral vertebrae provided do not only show unfused, but even slightly detached cervical ribs.
In summary, a much more detailed anatomical description and comparisons with other Morrison coelurosaurs are needed in this part of the manuscript.

For the phylogenetic analysis, the authors state that they evaluated all characters used in the different TWiG analyses over the years, reassessed them and recoded a vast number of taxa. This is certainly absolutely necessary and one of the strong aspects of the manuscript, for which the authors are to be commended! However, looking at the data matrix, I am somewhat unsure about the choice of including or excluding taxa from the analysis, and even though the authors briefly explain their rationale in the supplementary file, several aspects are left unclear.
One important aspect here would again be the phylogenetic comparison with other Morrison coelurosaurs. Of the three described Morrison coelurosaurs that are known from more than single elements (Ornitholestes, Coelurus, Tanycolagreus), only Ornitholestes is included in the matrix, and this taxon is used as outgroup, thus a priori excluding any closer relationship between the new specimen and this taxon. With the current taxon scope, the new specimen basically does not have any other choice but being a maniraptoran - which, as outlined above, is not obvious from the anatomical description.
Likewise, I am somewhat unsure why taxa such as Bahariasaurus (of which only short descriptions and some figures remain, and which was so far considered to be a non-coelurosaurian theropod by most authors) or Sciurumimus (described as a juvenile megalosaur) are included in an analysis focused on maniraptoriforms. The authors mention that they (as most iterations of the TWiG analysis) lack characters that would allow resolving the interrelationships of basal coelurosaurs, but as they would have to show that the new specimen actually is a maniraptoran rather than a more basal coelurosaur including such data would be crucial in this manuscript.

For more detailed comments please see the attached annotated manuscript.

Additional comments

In summary, I think this paper is potentially a very important addition to our knowledge of early coelurosaur evolution, if the authors provide a more detailed description and can clearly show that the specimen described is a new taxon and a maniraptoran (although even if this was only anew specimen of Coelurus or one of the other Morrison coelurosaurs, it has the potential to provide a wealth of new information on these generally rather poorly known taxa). I would thus strongly encourage the authors to do the necessary revisions and resubmit the manuscript!

---

## Round 0.2 · accepted · Accept

This will be a really important contribution to our understanding of non-avian theropod phylogeny, and I am pleased that you and your team chose PeerJ as your publication venue.

Reviewer 1 ·

Basic reporting

no comment

Experimental design

no comment

Validity of the findings

no comment

Additional comments

I thank the authors for entertaining my comments in the first round of review. I still have some reservations about the phylogenetic analysis and am concerned about the many heterodox results. However, I am not one to stand in the way of publication. I've said my piece, the authors have replied, and now it's important to get this new taxon and the dataset out there.